# Training Large Neural Networks With Low-Dimensional Error Feedback

## Abstract

Training deep neural networks typically relies on backpropagating high-dimensional error signals—a computationally intensive process with little evidence supporting its implementation in the brain. However, since most tasks involve low-dimensional outputs, we propose that low-dimensional error signals may suffice for effective learning. To test this hypothesis, we introduce a novel local learning rule based on Feedback Alignment that leverages indirect, low-dimensional error feedback to train large networks. Our method decouples the backward pass from the forward pass, enabling precise control over error signal dimensionality while maintaining high-dimensional representations. We begin with a detailed theoretical derivation for linear networks, which forms the foundation of our learning framework, and extend our approach to nonlinear and convolutional architectures. Remarkably, we demonstrate that even minimal error dimensionality—on the order of the task dimensionality—can achieve performance matching that of traditional backpropagation. Furthermore, our rule enables efficient training of convolutional networks, which have previously been resistant to Feedback Alignment methods, with minimal error. This breakthrough not only paves the way towards more biologically accurate models of learning but also challenges the conventional reliance on high-dimensional gradient signals in neural network training. Our findings suggest that low-dimensional error signals can be as effective as high-dimensional ones, prompting a reevaluation of gradient-based learning in high-dimensional systems. Ultimately, our work offers a fresh perspective on neural network optimization and contributes to understanding learning mechanisms in both artificial and biological systems.

## 1 Introduction

Neural networks, like the mind of a child, learn through whispers of correction—a subtle feedback that shapes their understanding of the world. Yet, while the world often communicates in the simplest of terms, our neural networks, with all their magnificent complexity, respond with an overabundance of noise.

Consider a typical real-world problem addressed by deep networks. Although the inputs to the networks are high-dimensional and the networks are overparameterized, the underlying structure of many real-world tasks is often far simpler—a low-dimensional core. Consequently, the error feedback received from the world, represented by the loss gradient, is also low-dimensional. Yet, as this error signal propagates backward through the network, it gains dimensionality, raising important questions about the efficiency and necessity of this approach.

The increase in error dimensionality occurs as the error signals propagate through the over-parameterized layers, a consequence of the inherent coupling between the feedforward and feedback processes. This coupling ensures that detailed error information is made available at each layer of the network, solving the credit-assignment problem. However, such high-dimensional error propagation is not strictly required for effective learning. In biological systems, particularly in the brain, error signals often travel through indirect and constrained pathways, hinting that lower-dimensional error feedback could suffice for learning. Despite this intriguing possibility, our understanding of how error signal dimensionality impacts learning—and whether similar principles operate in biological systems—remains limited.

Beyond the enigma of whether low-dimensional error signals suffice for effective training lies a deeper question: how do these signals shape the neural representations that emerge during learning? Specifically, we do not yet understand how the properties of error feedback influence neural tuning curves and receptive fields or how to connect these emergent features to their underlying learning mechanisms. Unraveling such connections could not only illuminate the processes by which the brain learns but also pave the way for more efficient learning strategies in artificial networks.

In this study, we investigate whether low-dimensional error pathways can serve as an effective alternative to training deep networks. We explore the hypothesis that low-dimensional error signals can drive efficient learning, thereby aligning artificial network training with the constraints seen in real-world tasks and biological systems. By extending current Feedback Alignment methods that decouple error feedback from the forward pass, we systematically manipulate the dimensionality of error signals, probing whether these constrained pathways can still support the rich and complex representations characteristic of deep learning.

## 2 BACKGROUND AND RELATED WORK

We consider a multilayerd perceptron with $L$ layers, each layer $l$ computes its output as $\boldsymbol{h}_l = f(W_l \boldsymbol{h}_{l-1})$, where $W_l$ is the weight matrix, and $f$ is an element-wise activation function. The input to the network is $\boldsymbol{h}_0 = \boldsymbol{x}$, and the final network output is $\hat{\boldsymbol{y}} = f_L(W_L \boldsymbol{h}_{L-1})$, which approximates the target $\boldsymbol{y}$. The *task dimensionality*, denoted $d$, is at most the number of components in $\boldsymbol{y}$ and $\hat{\boldsymbol{y}}$.

Training the network involves minimizing a loss function $\mathcal{L}(\boldsymbol{y}, \hat{\boldsymbol{y}})$ by adjusting the weights $\{W_l\}$. The error signal at the output layer, $\delta_L = \frac{\partial \mathcal{L}}{\partial \hat{\boldsymbol{y}}}$, is a $d$-dimensional vector, typically much smaller than the number of neurons in the hidden layers.

**Backpropagation (BP)** (Rumelhart et al., 1986) is the standard approach for training neural networks. It propagates the error backward through the network using $\delta_l = W_{l+1}^T \delta_{l+1} \odot f'(W_l \boldsymbol{h}_{l-1})$, and updates the weights using $\Delta W_l = -\eta \delta_l \boldsymbol{h}_{l-1}^T$, where $\eta$ is the learning rate. However, this method requires the exact transpose of the forward weights, $W_{l+1}^T$, which is biologically implausible (Grossberg, 1987; Crick, 1989). Moreover, BP tightly couples the error propagation with the forward pass, limiting the ability to explore how different properties of the error signal affect learning dynamics.

**Feedback Alignment (FA)** (Lillicrap et al., 2016) was proposed to address the biological limitations of BP by replacing $W_{l+1}^T$ with a fixed random matrix $B_l$. The error is computed as:

$$\delta_l = B_l \delta_{l+1} \odot f'(W_l \boldsymbol{h}_{l-1}), \tag{1}$$

decoupling the forward and backward weights and providing a more biologically plausible mechanism. However, FA struggles to scale effectively in deep architectures, such as convolutional neural networks (CNNs), where it often fails to achieve competitive performance (Bartunov et al., 2018).

An extension of FA involves adapting $B_l$ by updating it alongside the forward weights $W_l$ to improve their alignment (Kolen & Pollack, 1994; Akrout et al., 2019):

$$\Delta B_l = -\eta \boldsymbol{h}_{l-1} \delta_l^T - \lambda B_l, \quad \Delta W_l = -\eta \delta_l \boldsymbol{h}_{l-1}^T - \lambda W_l, \tag{2}$$

where $\lambda$ is a regularization parameter. Although this adaptive approach improves performance by better aligning forward and backward weights, it still requires high-dimensional error signals and struggles to match BP performance in complex architectures like CNNs. Furthermore, in Section 3, we show that this approach fails when the matrix $B$ is low-rank and the dimensionality of the error is constrained.

Other studies have explored the use of *fixed* sparse feedback matrices to reduce the dimensionality of error propagation (Crafton et al., 2019). However, these approaches result in significantly lower performance and do not provide a systematic framework for studying how error constraints affect learning and representation formation.

Beyond FA-based methods, several studies have shown that weight updates using backpropagation can result in a low-dimensional weight update Liao et al. (2016); Gunasekar et al. (2018); Caro et al. (2024) and favor low-rank solutions Patel & Shwartz-Ziv (2024). These findings support our hypothesis that a low-dimensional feedback is sufficient to train deep networks. However, no

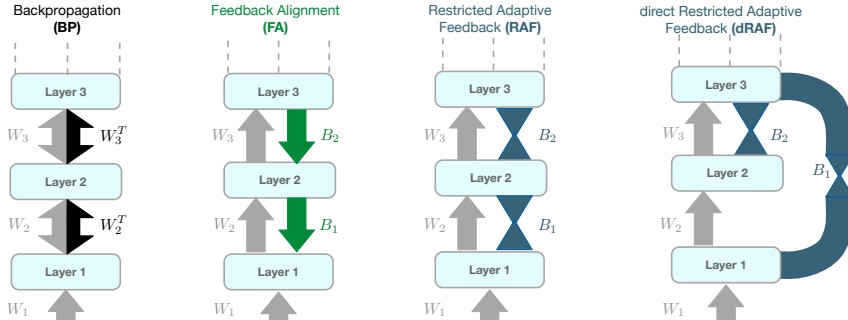

Figure 1: *Illustration of different approaches for propagating error to hidden layers.* From left to right: **BP** uses the transpose of the forward weights. In **FA**, error propagates through a fixed random matrix, aligning $\{W_l\}$ and $\{B_l\}$ over time to approximate BP. **RAF** uses low-rank feedback matrices, restricting error dimensionality and preventing a full mirroring of BP. **dRAF** extends RAF by allowing error propagation between non-consecutive layers or directly from the output layer.

previous work has considered training with a constrained error pathway, and the effects of error dimensionality and training have not been systematically studied.

In this work, our aim is to systematically investigate how constraining the dimensionality of the error signal affects the training and performance of neural networks. To this end, we introduce a novel learning scheme, *Restricted Adaptive Feedback (RAF)*, that allows flexible control over the dimensionality of the errors (Fig. 1).

**Our main contributions are**:

1. We present a novel learning rule, *Restricted Adaptive Feedback (RAF)*, which matches BP performance while requiring minimal error signals. We provide a detailed derivation of the learning dynamics in a simple linear case, establishing a foundational understanding of how RAF operates.

2. We demonstrate that nonlinear networks can efficiently learn nontrivial datasets using low-dimensional error signals, highlighting the versatility of RAF in practical scenarios.

3. We show that convolutional networks can also be effectively trained with low-dimensional feedback, addressing long-standing challenges in scaling learning biologically-inspired models.

4. We reveal that error dimensionality shapes the receptive fields in a model of the ventral visual system, offering new insights into the relationship between learning mechanisms and biological neural representations.

In the final section of this report, we discuss the broader implications of our results for both neuroscience and machine learning.

## 3 RESTRICTED ADAPTIVE FEEDBACK IN LINEAR NETWORKS

We begin our analysis by studying learning dynamics in multilayered linear networks. Although linear models may seem overly simplistic, they can exhibit rich learning dynamics due to the non-linearity introduced by the loss function (Saxe et al., 2013). Additionally, imposing dimensional constraints on linear networks yields insightful results that extend beyond the linear case.

**A linear problem** We consider a simple linear transformation problem with a low-dimensional structure, $\boldsymbol{y} = A\boldsymbol{x}$. Here, $\boldsymbol{x} \in \mathbb{R}^n$ represents the $n$-dimensional input, and $\boldsymbol{y} \in \mathbb{R}^m$ represents the target. The matrix $A$ is a rank-$d$ matrix defined as $A = \sum_{j=1}^d \boldsymbol{u}_j \boldsymbol{v}_j^T$, where $\boldsymbol{u}_j \in \mathbb{R}^n$ and $\boldsymbol{v}_j \in \mathbb{R}^m$

are random Gaussian vectors, and we assume $d \ll n$. Our data set consists of $p$ training samples $\{\boldsymbol{x}^\mu, \boldsymbol{y}^\mu\}_{\mu=1}^p$, with each input vector $\boldsymbol{x}^\mu$ being i.i.d. according to the standard normal distribution $\boldsymbol{x}^\mu \sim \mathcal{N}(\boldsymbol{0}, \boldsymbol{1})$. The labels are given by $\boldsymbol{y}^\mu = A\boldsymbol{x}^\mu + \boldsymbol{\xi}^\mu$, where $\boldsymbol{\xi}^\mu$ is additive Gaussian noise with zero mean and unit variance.

The goal is to learn the low-dimensional structure of $A$ from the $p$ samples using a linear neural network. For simplicity, we assume that $p$ is sufficiently large to allow the network to fully recover the structure of $A$.

**A linear network model.** To study the effects of restricted error pathways, we consider a simple linear network with three layers: an input layer $\boldsymbol{x} \in \mathbb{R}^n$, a hidden layer $\boldsymbol{h} \in \mathbb{R}^k$, and an output layer $\boldsymbol{y} \in \mathbb{R}^m$. The input and hidden layers are connected by the weight matrix $W_1 \in \mathbb{R}^{k \times n}$, and the hidden and output layers are connected by the weight matrix $W_2 \in \mathbb{R}^{m \times k}$. The output of the network can be expressed as $\boldsymbol{y} = W_2 W_1 \boldsymbol{x}$ (Fig. 2).

The network is trained to minimize the quadratic empirical loss function:

$$L = \frac{1}{p} \sum_\mu \|\boldsymbol{y}(\boldsymbol{x}^\mu) - W_2 W_1 \boldsymbol{x}^\mu\|^2. \tag{3}$$

We apply Feedback Alignment (FA) to update $W_1$, which does not have direct access to the loss gradient. Instead of backpropagating the error through $W_2^T$, we use a fixed low-rank feedback matrix $B$. This provides an alternative pathway for propagating the error signal to $W_1$.

For a given data point $\{\boldsymbol{x}^\mu, \boldsymbol{y}^\mu\}$, the weight updates, derived from the FA framework, are given by:

$$\Delta W_1^\mu = \eta B^T (\boldsymbol{y}^\mu - W_2 W_1 \boldsymbol{x}^\mu) \boldsymbol{x}^{\mu T}, \quad \Delta W_2^\mu = \eta (\boldsymbol{y}^\mu - W_2 W_1 \boldsymbol{x}^\mu) \boldsymbol{x}^{\mu T} W_1^T. \tag{4}$$

Here, $\eta$ represents the learning rate, and the update for $W_1$ is computed using the indirect error feedback provided by $B$, while $W_2$ receives the full error signal directly from the output.

**Constraining error dimensionality with low-rank feedback** To control the dimensionality of the error feedback, we impose a low-rank constraint on the feedback matrix $B$. Rather than allowing full-dimensional feedback, we decompose $B$ as $B = QP$, where $Q \in \mathbb{R}^{k \times r}$ and $P \in \mathbb{R}^{r \times m}$. When $r < \min(k, m)$, $B$ is low rank, which means that it can project the error signal onto at most $r$ independent directions.

This low-rank structure introduces an "$r$-bottleneck," which limits the flow of error information [1]. By controlling the value of $r$, we can systematically study how reducing the dimensionality of the error signal impacts learning. Initially, we follow the original Feedback Alignment framework, keeping $Q$ and $P$ as random matrices. However, as we will demonstrate, allowing $Q$ and $P$ to learn is crucial to high performance.

## 3.1 Learning Dynamics

Our analysis extends the framework established by Saxe et al. (2013) to incorporate indirect feedback with constrained dimensionality. We begin by characterizing the task across the $p$ data points through the input-output covariance matrix, $\Sigma_{io} = \frac{1}{p} \sum_{\mu=1}^p \boldsymbol{y}^\mu (\boldsymbol{x}^\mu)^T$, which captures the correlation between input vectors $\boldsymbol{x}$ and output vectors $\boldsymbol{y}$. Performing Singular Value Decomposition (SVD), we obtain $\Sigma_{io} = USV^T$, where $U \in \mathbb{R}^{m \times m}$ and $V \in \mathbb{R}^{n \times n}$ contain the left and right singular vectors, respectively, and $S \in \mathbb{R}^{m \times n}$ is a rectangular diagonal matrix of singular values. For sufficiently large $p$, the first $d$ singular values in $S$ correspond to the prominent directions in the data (i.e., the singular values of $A$), while the remaining singular values are $O(1/\sqrt{p})$ and dominated by noise.

To track how training aligns the network weights with these prominent directions, we rotate the weight matrices $W_1$, $W_2$, and $B$ using the singular vectors of $\Sigma_{io}$. This transformation simplifies

---

[1] In linear settings, the problem is solvable using a single weight matrix, rendering the training of $W_1$ unnecessary. However, this does not affect the learning dynamics. Furthermore, we can constrain the hidden layer size to be small ($k < d \log(n)$), rendering the learning the shallow weight ncessesary Johnson (1984).

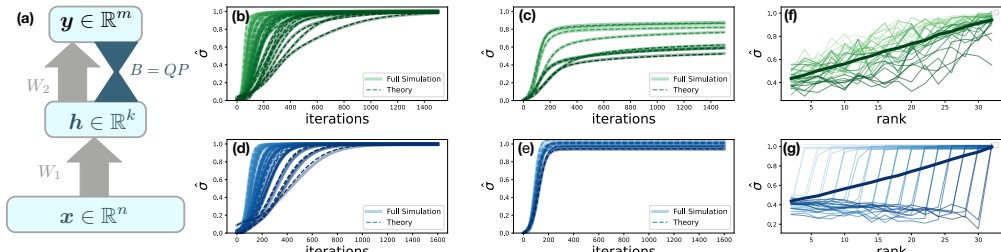

Figure 2: *Analysis of learning dynamics and component overlap in linear neural networks.* **(a)** Network architecture: $n = 128$, $k = 64$, $m = 64$, with backward matrices constrained to rank $r$. **(b,c)** Theoretical (dashed) vs. numerical (solid) dynamics for low-rank FA, training only $Q$ ($r = 64$ and $r = 8$, respectively). The $y$-axis shows the singular vectors overlap between $W_2 W_1$ and $\Sigma_{io}$. **(d,e)** Same as (b,c), but with RAF, training both $Q$ and $P$. **(f)** For $r < d$, without training $P$ the singular modes are learned on average (bold), but the top $r$ do not. **(g)** In RAF, simlar average behavior (bold) but the top $r$ components are fully recovered.

the analysis by aligning the network's weight dynamics with the key data directions:

$$W_1 = \bar{W}_1 V^T, \quad W_2 = U \bar{W}_2, \quad B = \bar{B} U^T,$$

where $\bar{W}_1$, $\bar{W}_2$, and $\bar{B}$ represent the transformed weight matrices. This rotation aligns the weight dynamics with the dominant singular vectors, allowing us to focus on how the network captures the important features of the data.

Since the inputs are uncorrelated, we can apply these transformations to the iterative weight-update equations derived from the FA learning rule. Assuming a small learning rate $\eta \ll 1$ with full-batch updates, we express the weight updates in continuous time:

$$\tau \frac{d\bar{W}_1}{dt} = \bar{B}^T (S - \bar{W}_2 \bar{W}_1), \quad \tau \frac{d\bar{W}_2}{dt} = (S - \bar{W}_2 \bar{W}_1) \bar{W}_1^T, \tag{5}$$

where $\eta = dt/\tau$. This continuous form captures the dynamics of the learning process, allowing us to study it from a dynamical systems perspective. By analyzing these equations, we can identify fixed points and evaluate their stability, providing insight into how the network converges and learns under constrained feedback. Fig. 2, shows how the singular vectors of $W_2 W_1$ align to the corresponding singular vectors of $\Sigma_{io}$.

**Stationary solutions for the training.** Training halts when the right-hand side of the weight update equations (5) vanishes, indicating that the dynamics have reached a stable fixed point. At this fixed point, the update equation for $\bar{W}_1$ leads to the condition:

$$\bar{B}(S - \bar{W}_2 \bar{W}_1) = 0 \implies S_{jj} \boldsymbol{B}_{:,j} = \sum_{i=1}^{m} \boldsymbol{B}_{:,i} (\bar{W}_2 \bar{W}_1)_{ij} \quad \forall j, \tag{6}$$

where $\boldsymbol{B}_{:,j} \in \mathbb{R}^k$ is the $j$-th column of $\bar{B}$, and $S_{jj}$ is the $j$-th singular value of $\Sigma_{io}$. This equation indicates that, at the stationary point, the weight products $\bar{W}_2 \bar{W}_1$ must align with the singular modes of the data.

However, since $B$ is of rank $r$, the feedback matrix $\bar{B}$ can only span at most $r$ independent directions. If $r = m$, the system has enough feedback dimensionality to align perfectly with the singular values in $S$, recovering the full structure of $\Sigma_{io}$ as demonstrated in Saxe et al. (2013). In this case, the training successfully converges to a unique solution where $\bar{W}_2 \bar{W}_1 = S$ (Fig. 2b).

Crucially, when $r < m$, the feedback matrix $\bar{B}$ lacks the sufficient rank to fully capture the $m$ independent directions in $S$. As a result, eq. (6) becomes under-determined, leading to potentially infinite solutions. This means that the trained weights $\bar{W}_2 \bar{W}_1$ may not align with the true data structure encoded in $\Sigma_{io}$ (Fig. 2c)

These findings reveal that, in the case of low-rank feedback, Feedback Alignment (FA) is insufficient to ensure convergence to the correct solution. This necessitates training the feedback matrix to align properly with the data, ensuring that the network learns the correct representations.

## 3.2 TRAINING THE FEEDBACK WEIGHTS

From the previous analysis, it is apparent that the feedback matrix $B$ must be trained for the learning dynamics to converge to the correct solution of eq. (6). When $B$ is low-rank and fixed, the network lacks sufficient capacity to transmit the full error information, which can impede learning.

One approach is to adopt a learning rule inspired by Kolen-Pollack, as in eq. (2). When $B = QP$, the updates are given by

$$\Delta W_2^\mu = \eta \boldsymbol{\delta}^\mu \boldsymbol{x}^{\mu T} W_1^T - \lambda W_2 \quad \text{and} \quad \Delta Q^\mu = \eta W_1 \boldsymbol{x}^\mu \boldsymbol{\delta}^{\mu T} P^T - \lambda Q, \tag{7}$$

where $\boldsymbol{\delta}^\mu = \frac{\partial \mathcal{L}(\boldsymbol{x}^\mu, \boldsymbol{y}^\mu)}{\partial \boldsymbol{y}} = \boldsymbol{y}^\mu - W_2 W_1 \boldsymbol{x}^\mu$ is the error gradient for the $\mu$-th data point, $\eta$ is the learning rate, and $\lambda$ is the regularization parameter. Importantly, in this framework only the column space of $B$, or in our case $Q$ are updated.

However, adopting this learning framework is insufficient when $P$ remains fixed and randomly initialized (Fig. 2c). Since $P$ can project onto at most $r$ unique directions of the error, it may not align with the relevant error subspace. In this case, the network may converge to an incorrect solution, as indicated by the non-uniqueness of solutions to (6) when $m > r$.

Ideally, we want $P$ to be an orthogonal matrix whose $r$ columns span the top $r$ principal directions of the output-output correlation matrix $\Sigma^{yy}$. This alignment ensures that the most significant components of the error are propagated back through the network.

In cases where the output correlations are unknown, we can update $P$ using a modified Oja learning rule Oja (1982):

$$\Delta P^\mu = \eta P \boldsymbol{y}^\mu \boldsymbol{y}^{\mu T}(I - P^T P) - \lambda P, \tag{8}$$

where $I$ is the identity matrix. This rule adjusts $P$ incrementally so that its columns converge to the top $r$ principal components of the outputs $\{\boldsymbol{y}^\mu\}$.

By training both $Q$ and $P$, we allow the feedback matrix $B = QP$ to adaptively align with the relevant error directions, enabling the network to learn the correct mappings even under constrained feedback dimensionality.

Repeating the linear analysis that led to (5), we extend the derivation to our case with adaptive feedback weights. We define the transformed feedback matrix as $P = \bar{P}U^T$. This transformation aligns the feedback matrix $P$ with the principal components of the data, simplifying the analysis.

Taking the continuous-time limit (with $\eta \to 0$ and $\eta p = \tau$), we obtain a set of differential equations that describe the learning dynamics of the forward and backward weights:

$$\tau \frac{d\bar{W}_1}{dt} = \bar{B}^T(S - \bar{W}_2 \bar{W}_1), \qquad \tau \frac{dQ}{dt} = \bar{W}_1(S - \bar{W}_2 \bar{W}_1)^T \bar{P}^T - \lambda Q,$$
$$\text{and} \tag{9}$$
$$\tau \frac{d\bar{W}_2}{dt} = (S - \bar{W}_2 \bar{W}_1)\bar{W}_1^T - \lambda \bar{W}_2, \qquad \tau \frac{d\bar{P}}{dt} = \bar{P}SS^T(I - \bar{P}^T \bar{P}) - \lambda \bar{P}.$$

Here, $\bar{B} = Q\bar{P}$ and $\lambda$ is the regularization parameter. Note that $Q$ is not affected by rotation, as it does not come in contact with either the input or the output. The full derivation can be found in the Appendix.

Notably, the updates to the feedback weights are local and follow learning rules that were well-studied in theoretical neuroscience Oja (1982); Clopath et al. (2010); Turrigiano (2008); Pehlevan et al. (2015). Local plasticity makes our framework an attractive alternative for backpropagation in models of brain circuits.

We refer to this learning framework as *Restricted Adaptive Feedback* (RAF). Fig. 2 compares the learning dynamics of RAF with those of BP and FA where only the $Q$ matrix is learned. The results demonstrate that RAF effectively aligns the feedback weights, enabling the network to converge to the correct solution despite the constrained error dimensionality. Furthermore, RAF will learn the top $r$ components of $\Sigma$, even if $r < d$ (Fig. 2e).

## 3.3 RESTRICTED ADAPTIVE FEEDBACK IN DEEP ARCHITECTURES

Our weight-update equations can be naturally extended to deeper networks. In the single hidden layer model above, the backward weights $P$ were updated using the true labels $\boldsymbol{y}^\mu$. However, in

deeper models, hidden layers do not have ground-truth representations. Instead, each layer relies on the local error signal, which propagates through the network according to $\boldsymbol{\delta}_l^\mu = Q_l P_l \boldsymbol{\delta}_{l+1}^\mu$. This error signal $\boldsymbol{\delta}_l^\mu$ provides the necessary information for learning at layer $l$.

To update the feedback weights $P_l$ in the absence of ground-truth representations, we use local error signals $\boldsymbol{\delta}_{l+1}^\mu$. As in the single-layer model, we use Oja's rule to adjust $P_l$ to span the principal components of the error at the next layer, $\boldsymbol{\delta}_{l+1}^\mu$, ensuring efficient error propagation.

The complete update rules for layer $l$ are given by

$$\Delta W_l^\mu = \eta Q_l P_l \boldsymbol{\delta}_{l+1}^\mu \boldsymbol{h}_l^{\mu T} - \lambda W_l \quad \text{and} \quad \begin{aligned} \Delta Q_l^\mu &= \eta \boldsymbol{h}_l^\mu P_l \boldsymbol{\delta}_{l+1}^\mu - \lambda Q_l, \\ \Delta P_l^\mu &= \eta P_l \boldsymbol{\delta}_{l+1}^\mu \boldsymbol{\delta}_{l+1}^{\mu T}(I - P_l^T P_l) - \lambda P, \end{aligned} \tag{10}$$

where $\eta$ is the learning rate, $\lambda$ is the regularization parameter, $\boldsymbol{h}_l^\mu$ is the activation of layer $l$, and $\boldsymbol{\delta}_{l+1}^\mu$ is the error signal from the next layer.

By updating $P_l$ using the error signals, we ensure that the feedback weights of each layer are adapted to capture the most relevant directions in the error space, facilitating effective learning throughout the network. Notably, while we use the same learning rate $\eta$ and weight decay $\lambda$ for all components $\{W_l\}$, $\{Q_l\}$, and $\{P_l\}$, it can potentially differ.

In the interest of brevity, we omit simulations of deep linear networks, as the extension from the single-layer case is straightforward. Instead, we proceed directly to deep nonlinear networks, where the impact of constrained error feedback presents more complex and interesting dynamics.

## 4 MINIMAL ERROR IS SUFFICIENT TO TRAIN NONLINEAR NETWORKS

Adapting our Restricted Adaptive Feedback (RAF) framework to nonlinear networks is straightforward because the core principles of local learning and constrained error feedback remain applicable. The local update rules in eq. (10) remain the same; the primary difference lies in the introduction of nonlinear activation functions during the propagation of signals and errors. Specifically, the forward and backward passes are modified as follows:

$$h_l = f(W_l h_{l-1}), \quad \delta_l = Q_l P_l \delta_{l+1} \odot f'(h_l), \tag{11}$$

where $f$ is the nonlinear activation function applied element-wise, and $f'$ is its derivative.

To test whether our learning rule extends effectively from linear to nonlinear models, we trained deep networks on the CIFAR-10 dataset. We used a simple nonlinear model with four fully connected layers of 512 ReLU neurons each. While not state-of-the-art, this model provides a suitable testbed for evaluating our theory's applicability to nonlinear architectures and complex data.

To isolate the impact of feedback dimensionality, we applied the Restricted Adaptive Feedback (RAF) rule in eq. (10), constraining the rank of feedback matrices in one specific layer at a time while leaving the others unrestricted. We then measured the network's test accuracy and compared it to backpropagation (BP) as a baseline. Fig. 3a shows the accuracy as a function of the feedback rank $r_l$, with each curve representing a different constrained layer.

Consistent with our findings for the linear model, constraining the feedback rank to $r = d = 10$, in any layer, match BP performance (Fig. 3a), where $d$ is the number of classes. Interestingly, the shallower layers performed well even under tighter rank constraints ($r_l < d$), suggesting that the deeper layers compensate for the limited feedback in the earlier layers by effectively adjusting their weights.

This compensatory effect is possible because no information is lost during the feedforward pass, unlike in a bottlenecked network. To demonstrate that the network still utilizes high-dimensional representations, we compared the performance of RAF-trained networks with constrained feedback to narrower networks without rank restrictions (Fig. 3b). The results confirm that RAF-trained networks leverage their width to maintain high performance despite feedback constraints.

Moreover, as shown in Fig. 3b, nonlinear networks trained with low-rank feedback in all layers using RAF can still match BP performance, demonstrating RAF's robustness even with dimensional constraints across the entire network.

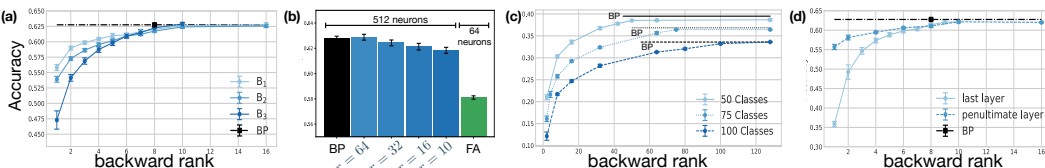

Figure 3: *Restricted Adaptive Feedback (RAF) efficiently trains nonlinear networks on CIFAR-10.* (a) Restricting feedback in any layer to $r = d = 10$ matches full BP performance. (b) Restricting feedback does not significantly reduce performance, while reducing network width does, indicating the use of high-dimensional representations. (c) Minimal error dimensionality matches task dimensionality, shown by subsampling classes from CIFAR-100. All feedback matrices constrained to $r$. (d) Direct RAF: all layers receive error from the output (solid) or penultimate layer (dashed), converging to BP performance.

**Task Dimensionality Determines the Minimal Rank** Our linear analysis suggests that the error signal dimensionality needed for effective learning is tied to the loss gradient dimensionality, which depends on the number of classes in the data. To test this, we trained networks on subsets of CIFAR-100 with 50, 75, and 100 classes, constraining the ranks of all feedback matrices to $r$ (Fig. 3c).

The results show that network performance matches BP when the feedback rank equals the number of classes ($r = d$). This indicates that the minimal rank required for effective learning aligns with the task's complexity, as defined by the number of output classes.

**Direct Restricted Adaptive Feedback (dRAF)** Our theory shows how to propagate error signals from deeper layers to shallower ones using restricted adaptive feedback. However, error projections can also bypass intermediate layers entirely, leading to different variants of *Direct Restricted Adaptive Feedback* (dRAF), analogous to Direct Feedback Alignment (Nøkland, 2016). For example, we can make direct connections from the output or the penultimate layer to earlier layers, training these connections using our algorithm (Fig. 3d). As with RAF, this direct projection method matches the performance of BP when the rank of the feedback matrices satisfies $r \geq d$. This model is particularly important because it is more flexible and has greater potential to explain learning in the brain, where error signals may arrive from different pathways.

## 4.1 Convolutional Neural Networks

Our previous results demonstrate that fully connected networks can learn effectively from minimal error signals, even on complex datasets, matching backpropagation (BP) performance. Here, we extend this investigation to convolutional architectures.

Training convolutional networks with Feedback Alignment (FA) is notoriously difficult (Bartunov et al., 2018; Launay et al., 2019). Recent work has made progress by learning feedback weights within the FA framework (Bacho & Chu, 2024), but our approach differs by restricting the error signal's dimensionality using Restricted Adaptive Feedback (RAF). We aim to determine whether convolutional networks can also benefit from low-dimensional error feedback.

We trained a VGG-like convolutional network with four blocks and batch normalization on the CIFAR-10 dataset (see Appendix for details). Using RAF, we decoupled the error propagation from the feedforward pass in all layers. Initially, we constrained the feedback error only in the blocks containing 512 (Fig. 4a). Consistent with our findings in fully connected networks, the convolutional networks learn well with a feedback matrix with a rank similar to the number of classes, $d = 10$.

To test whether convolutional networks can train when constraining *all* feedback paths, we further constrained each block to have feedback matrices with ranks equal to $1/2$, $1/4$, or $1/8$ of the block width (Fig. 4b). Our results indicate that reducing the error dimensionality has minimal impact on performance, except in the most extreme case. Specifically, constraining the feedback rank to $1/8$ of the block width resulted in a noticeable drop in performance. This finding aligns with our previous results, as the layer with 64 channels received feedback with a rank of $r = 8$. Overall, our results demonstrate that convolutional networks can be efficiently trained using a minimal error signal.

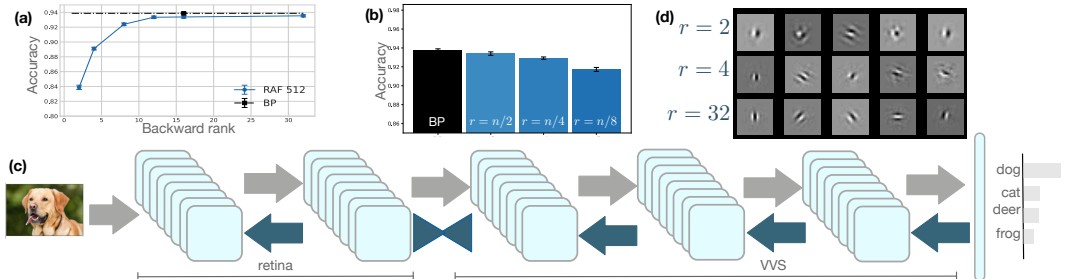

Figure 4: *Convolutional networks and receptive fields.* **(a)** Performance of a 4-block VGG-like network trained with RAF, constraining the layers with 512 channels. **(b)** Training the same network as in (a), with all layers constrained to a fraction of their size; the smallest layer has 64 channels. **(c)** Model of the early ventral visual stream, adapted from Lindsey et al. (2019). **(d)** Top receptive fields of (c). Constrained feedback to the retina result in more center-surround fields.

## 5 ERROR DIMENSIONALITY SHAPES NEURAL RECEPTIVE FIELDS

We have shown that neural networks can be efficiently trained using minimal error signals comparable to task dimensionality. Here, we investigate how error dimensionality affects neural representations, providing insights into receptive fields observed in the brain.

Lindsey et al. (2019) found that in convolutional models of the visual system, narrow feedforward bottlenecks between the retina and the brain led to center-surround receptive fields in the retinal layer, similar to mammals. Wider bottlenecks resulted in orientation-selective receptive fields, as seen in salamanders. We hypothesize that these effects are due to constraints on the error signal reaching the retina, rather than the feedforward bottlenecks themselves.

To test this hypothesis, we trained a model similar to that of Lindsey et al. (2019) but with full-width layers throughout. Instead of constraining the feedforward pathway, we constrained only the error signal using a low-rank feedback matrix trained with RAF (Fig. 4c), thereby isolating the impact of error dimensionality on neural representations. We extracted the receptive fields of neurons in the retinal layer using visualization techniques (Erhan et al., 2009) (Fig. 4d).

Consistent with our hypothesis, constraining the feedback rank led to the emergence of center-surround receptive fields in the retinal layer. To further validate our hypothesis, we trained a model that included the feedforward bottleneck, similar to Lindsey et al. (2019) but used dRAF to train the retinal layer without restricting the backward pathway. In line with our expectations, the retinal receptive fields exhibited orientation selectivity (see Appendix).

This experiment demonstrates that error dimensionality influences neuronal tuning and neural representations. Specifically, lower-dimensional error signals promote higher symmetries in emergent receptive fields. Our findings underscore the importance of considering the dimensionality and pathways of error signals when studying neural computations in the brain.

## 6 DISCUSSION

Our work demonstrates that neural networks can be trained effectively using minimal error signals constrained to the task's intrinsic dimensionality, rather than the higher dimensionality of the network's representations. By adopting a factorized version of Feedback Alignment with low-rank matrices and training both left and right spaces of the feedback matrix we showed that deep networks—linear, nonlinear, and convolutional—can achieve performance comparable to full backpropagation even under stringent error-dimensionality constraints. This finding highlights that the essential information required for learning is tied to the complexity of the task, as measured by the number of output classes. Additionally, we revealed that constraining error dimensionality influences neural representations, providing a potential explanation for biological phenomena, such as center-surround receptive fields in the retina.

The main goal of this work is to explore potential mechanisms for implementing gradient descent in the brain. Recognizing that high-dimensional feedback is not necessary for effective learning is a significant step toward developing more flexible and biologically realistic models of learning. This insight suggests that the brain might utilize low-dimensional error signals to drive learning processes, aligning with the anatomical and physiological constraints.

While our novel learning rule bears superficial similarity to previous Feedback Alignment (FA) schemes, it is *conceptually different*. In traditional FA, learning the feedback weights aims to align them with the feedforward weights Lillicrap et al. (2016); Akrout et al. (2019), mirroring full back-propagation. In contrast, by factorizing the feedback matrix as $B = QP$ we also *align the feedback's row space with the source of the error*, thereby improving the quality of the error signal itself. This approach not only enhances learning efficiency but may also act as a form of regularization, a possibility that warrants further investigation.

Our findings invite a rethinking of gradient descent dynamics in overparameterized networks. Typically, the weight dynamics during training are high-dimensional. However, when the error signal is low-dimensional, the weight updates in each layer are confined to a much lower-dimensional subspace. This constraint could have implications for understanding the generalization capabilities of neural networks, as it suggests that effective learning does not require exploring the full parameter space. Exploring this connection could open new avenues for understanding the dynamics of gradient descent in high-dimensional loss spaces.

In summary, our work highlights the critical role of error signal dimensionality in learning and representation formation within neural networks. By demonstrating that low-dimensional error feedback is sufficient for effective training, we bridge a gap between artificial neural network training and biological limitations. This alignment advances our understanding of how the brain implements supervised learning and provides a foundation for extending current learning frameworks.

## REPRODUCIBILITY STATEMENT

To ensure the reproducibility of our results, we have provided detailed derivations of all theoretical results in the Appendix, along with comprehensive descriptions of the network architectures, datasets, and training algorithms used in our experiments. Additionally, upon publication, we will make the complete codebase available on GitHub, including the exact scripts used to generate all figures presented in this paper.

## ETHICS STATEMENT

This work is purely theoretical, focusing on the development of new learning algorithms and their implications for artificial and biological learning systems. No human or animal subjects were involved in this research. We used only freely available datasets (such as CIFAR-10 and CIFAR-100) that are standard in the machine learning community and do not contain personally identifiable information. We do not foresee any direct ethical concerns related to the outcomes or applications of this work.

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

APPENDIX

## A  LINEAR THEORY FOR FEEDBACK ALIGNMENT

From eq. (4) in the main text, the weight updates for a single sample $\mu$ are given by:

$$
\begin{aligned}
\Delta W_1^\mu &= \eta B \left( \boldsymbol{y}^\mu - W_2 W_1 \boldsymbol{x}^\mu \right) \boldsymbol{x}^{\mu\top}, \\
\Delta W_2^\mu &= \eta \left( \boldsymbol{y}^\mu - W_2 W_1 \boldsymbol{x}^\mu \right) \boldsymbol{x}^{\mu\top} W_1^\top,
\end{aligned}
\tag{12}
$$

where:

- $\eta$ is the learning rate,
- $\boldsymbol{x}^\mu \in \mathbb{R}^n$ is the input vector for sample $\mu$,
- $\boldsymbol{y}^\mu \in \mathbb{R}^m$ is the corresponding target output,
- $W_1 \in \mathbb{R}^{k \times n}$ and $W_2 \in \mathbb{R}^{m \times k}$ are the weight matrices,
- $B \in \mathbb{R}^{k \times m}$ is a predefined matrix (e.g., a feedback or scaling matrix).

We introduce the empirical covariance matrices:

$$
\Sigma_{io} = \frac{1}{p} \sum_{\mu=1}^{p} \boldsymbol{y}^\mu \boldsymbol{x}^{\mu\top},
\tag{13}
$$

$$
\Sigma_{oo} = \frac{1}{p} \sum_{\mu=1}^{p} \boldsymbol{y}^\mu \boldsymbol{y}^{\mu\top},
\tag{14}
$$

$$
\Sigma_{ii} = \frac{1}{p} \sum_{\mu=1}^{p} \boldsymbol{x}^\mu \boldsymbol{x}^{\mu\top} = I,
\tag{15}
$$

where $\Sigma_{ii} = I$ assumes that the input vectors are whitened (i.e., have unit covariance). Summing over all $p$ training examples, we obtain the average weight updates:

$$
\begin{aligned}
\Delta W_1 &= \frac{\eta}{p} \sum_{\mu=1}^{p} \Delta W_1^\mu \\
&= \frac{\eta}{p} \sum_{\mu=1}^{p} B \left( \boldsymbol{y}^\mu - W_2 W_1 \boldsymbol{x}^\mu \right) \boldsymbol{x}^{\mu\top}.
\end{aligned}
\tag{16}
$$

Using these definitions, the update for $W_1$ simplifies to:

$$
\Delta W_1 = \eta B \left( \Sigma_{io} - W_2 W_1 \Sigma_{ii} \right)
\tag{17}
$$

$$
= \eta B \left( \Sigma_{io} - W_2 W_1 \right).
\tag{18}
$$

Under the limit as $\eta \to 0$ with $\eta = \frac{dt}{\tau}$ (where $\tau$ is a time constant), we transition from discrete updates to continuous-time dynamics:

$$
\tau \frac{dW_1}{dt} = B \left( \Sigma_{io} - W_2 W_1 \right),
\tag{19}
$$

which matches the weight dynamics presented in eq. (5) of the main text.

Similarly, the update for $W_2$ becomes:

$$
\Delta W_2 = \frac{\eta}{p} \sum_{\mu=1}^{p} \Delta W_2^\mu
\tag{20}
$$

$$
= \frac{\eta}{p} \sum_{\mu=1}^{p} \left( \boldsymbol{y}^\mu - W_2 W_1 \boldsymbol{x}^\mu \right) \boldsymbol{x}^{\mu\top} W_1^\top
\tag{21}
$$

$$
= \eta \left( \Sigma_{io} - W_2 W_1 \right) W_1^\top,
\tag{22}
$$

which simplifies under the same limit to:

$$\tau \frac{dW_2}{dt} = \left(\Sigma_{io} - W_2 W_1\right) W_1^\top. \tag{23}$$

Eq. (19) and eq. (23) describe the continuous-time dynamics of the weights $W_1$ and $W_2$ under the given learning rule.

We consider the singular value decomposition (SVD) of the covariance matrix $\Sigma_{io}$:

$$\Sigma_{io} = USV^\top, \tag{24}$$

where:

- $U \in \mathbb{R}^{m \times d}$ and $V \in \mathbb{R}^{n \times d}$ are matrices with orthonormal columns,
- $S \in \mathbb{R}^{d \times d}$ is a diagonal matrix containing the singular values,
- $d$ is the rank of $\Sigma_{io}$.

We perform a rotation of the weight matrices and $B$ as follows:

$$W_1 = \bar{W}_1 V^\top,$$
$$W_2 = U \bar{W}_2, \tag{25}$$
$$B = \bar{B} U^\top.$$

Substituting these into the previous weight dynamics, we have for $W_1$:

$$\tau \frac{dW_1}{dt} = B \left(\Sigma_{io} - W_2 W_1\right) \tag{26}$$
$$= \bar{B} U^\top \left(USV^\top - U\bar{W}_2 \bar{W}_1 V^\top\right). \tag{27}$$

Since $U^\top U = I$ (due to orthonormal columns of $U$), we can simplify:

$$\tau \frac{dW_1}{dt} = \bar{B} \left(U^\top U\right) \left(S - \bar{W}_2 \bar{W}_1\right) V^\top \tag{28}$$
$$= \bar{B} \left(S - \bar{W}_2 \bar{W}_1\right) V^\top. \tag{29}$$

Recognizing that $W_1 = \bar{W}_1 V^\top$, we can write $\frac{dW_1}{dt} = \frac{d\bar{W}_1}{dt} V^\top$. Thus, multiplying both sides on the right by $V$ (since $V^\top V = I$):

$$\tau \frac{d\bar{W}_1}{dt} = \bar{B} \left(S - \bar{W}_2 \bar{W}_1\right). \tag{30}$$

Similarly, for $W_2$:

$$\tau \frac{dW_2}{dt} = \left(\Sigma_{io} - W_2 W_1\right) W_1^\top \tag{31}$$
$$= \left(USV\top - U\bar{W}_2 \bar{W}_1 V^\top\right) \left(\bar{W}_1 V^\top\right)^\top \tag{32}$$
$$\tag{33}$$

Using the fact that $V^\top V = I$ we simplify:

$$\tau \frac{dW_2}{dt} = \left(US - U\bar{W}_2 \bar{W}_1\right) \bar{W}_1^\top \tag{34}$$
$$\tag{35}$$

Since $W_2 = U\bar{W}_2$, we have $\frac{dW_2}{dt} = U \frac{d\bar{W}_2}{dt}$. Multiplying both sides on the left by $U^\top$:

$$\tau \frac{d\bar{W}_2}{dt} = \left(S - \bar{W}_2 \bar{W}_1\right) \bar{W}_1^\top. \tag{36}$$

Eqs. (30) and (36) describe the dynamics of the rotated weights $\bar{W}_1$ and $\bar{W}_2$:

$$\tau \frac{d\bar{W}_1}{dt} = \bar{B} \left(S - \bar{W}_2 \bar{W}_1\right),$$
$$\tau \frac{d\bar{W}_2}{dt} = \left(S - \bar{W}_2 \bar{W}_1\right) \bar{W}_1^\top. \tag{37}$$

## B    LINEAR THEORY FOR RAF

We consider an alternative algorithm where the matrix $B$ is replaced by $B = QP$, with $Q \in \mathbb{R}^{k \times r}$ and $P \in \mathbb{R}^{r \times m}$.

The updates for $Q$ and $P$ for a single sample $\mu$ are given by:

$$\Delta Q^\mu = \eta W_1 \boldsymbol{x}^\mu P \left(\boldsymbol{y}^\mu - W_2 W_1 \boldsymbol{x}^\mu\right),$$
$$\Delta P^\mu = \eta P \boldsymbol{y}^\mu \boldsymbol{y}^{\mu\top} \left(I - P^\top P\right). \tag{38}$$

Summing over all $p$ training examples, we obtain the average updates:

$$\Delta Q = \frac{\eta}{p} \sum_{\mu=1}^{p} \Delta Q^\mu = \eta W_1 \left(\frac{1}{p} \sum_{\mu=1}^{p} \boldsymbol{x}^\mu \left(\boldsymbol{y}^\mu - W_2 W_1 \boldsymbol{x}^\mu\right)^\top P^\top\right),$$

$$\Delta P = \frac{\eta}{p} \sum_{\mu=1}^{p} \Delta P^\mu = \eta P \left(\frac{1}{p} \sum_{\mu=1}^{p} \boldsymbol{y}^\mu \boldsymbol{y}^{\mu\top}\right) \left(I - P^\top P\right). \tag{39}$$

We simplify the update for $Q$:

$$\Delta Q = \eta W_1 \left(\left(\frac{1}{p} \sum_{\mu=1}^{p} \boldsymbol{x}^\mu \left(\boldsymbol{y}^\mu - W_2 W_1 \boldsymbol{x}^\mu\right)^\top\right) P^\top\right) \tag{40}$$

$$= \eta W_1 \left(\left(\Sigma_{io}^\top - W_1^\top W_2^\top \Sigma_{ii}\right) P^\top\right) \tag{41}$$

$$= \eta W_1 \left(\left(\Sigma_{io} - W_2 W_1\right)^\top P^\top\right) \tag{42}$$

Similarly, the update for $P$ simplifies to:

$$\Delta P = \eta P \Sigma_{oo} \left(I - P^\top P\right). \tag{43}$$

Thus, the updates for $Q$ and $P$ become:

$$\Delta Q = \eta W_1 \left(\Sigma_{io} - W_2 W_1\right)^\top P^\top,$$
$$\Delta P = \eta P \Sigma_{oo} \left(I - P^\top P\right). \tag{44}$$

Under the continuous-time assumption, where $\eta \to 0$ with $\eta = \frac{dt}{\tau}$, the updates for $Q$ and $P$ become differential equations:

$$\tau \frac{dQ}{dt} = W_1 \left(\Sigma_{io} - W_2 W_1\right)^\top P^\top, \tag{45}$$

and

$$\tau \frac{dP}{dt} = P \Sigma_{oo} \left(I - P^\top P\right). \tag{46}$$

Finally, substituting $B = QP$ into the update for $W_1$ from eq. 19, we find that the dynamics for $W_1$ become:

$$\tau \frac{dW_1}{dt} = QP \left(\Sigma_{io} - W_2 W_1\right) \tag{47}$$

We perform rotations similar to before:

$$W_1 = \bar{W}_1 V^\top,$$
$$W_2 = U \bar{W}_2, \tag{48}$$
$$P = \bar{P} U^\top.$$

Substituting for $Q$ we get:

$$\tau \frac{dQ}{dt} = \bar{W}_1 V_1^\top \left(USV^\top - U\bar{W}_2\bar{W}_1 V^\top\right)^\top \left(PU^\top\right)^\top \tag{49}$$

$$= \bar{W}_1 V_1^\top V \left(S - \bar{W}_2\bar{W}_1\right)^T U^\top U \bar{P}^\top \tag{50}$$

$$= \bar{W}_1 \left(S - \bar{W}_2\bar{W}_1\right)^\top \bar{P}^\top \tag{51}$$

Substituting these rotations into the previous derivations, we update the dynamics.

For $W_1$, the update equation is:

$$\tau \frac{dW_1}{dt} = B \left( \Sigma_{io} - W_2 W_1 \right).$$

(52)

Since $B = QP = Q\bar{P}U^\top$, $W_1 = \bar{W}_1 V^\top$, $W_2 = U\bar{W}_2$, and $\Sigma_{io} = USV^\top$, we have:

$$\tau \frac{d(\bar{W}_1 V^\top)}{dt} = Q\bar{P}U^\top \left( USV^\top - U\bar{W}_2 \bar{W}_1 V^\top \right)$$

(53)

$$= Q\bar{P} \left( SV^\top - \bar{W}_2 \bar{W}_1 V^\top \right).$$

(54)

Since $V^\top$ is constant, we can write:

$$\tau \frac{d\bar{W}_1}{dt} V^\top = Q\bar{P} \left( SV^\top - \bar{W}_2 \bar{W}_1 V^\top \right).$$

(55)

Multiplying both sides on the right by $V$ (using $V^\top V = I$):

$$\tau \frac{d\bar{W}_1}{dt} = Q\bar{P} \left( S - \bar{W}_2 \bar{W}_1 \right).$$

(56)

Substituting $P = \bar{P}U^\top$ and $\Sigma_{oo} = US^2 U^\top$ in (46), we get:

$$\tau \frac{d(\bar{P}U^\top)}{dt} = \bar{P}U^\top \left( US^2 U^\top \right) \left( I - U\bar{P}^\top \bar{P}U^\top \right)$$

(57)

$$= \bar{P}S^2 \left( I - \bar{P}^\top \bar{P} \right) U^\top.$$

(58)

Multiplying both sides on the right by $U$ (since $U^\top U = I$):

$$\tau \frac{d\bar{P}}{dt} = \bar{P}S^2 \left( I - \bar{P}^\top \bar{P} \right).$$

(59)

In summary, under the rotations, the updated dynamics are:

$$\begin{aligned}
\tau \frac{d\bar{W}_1}{dt} &= Q\bar{P} \left( S - \bar{W}_2 \bar{W}_1 \right), \\
\tau \frac{d\bar{W}_2}{dt} &= \left( S - \bar{W}_2 \bar{W}_1 \right) \bar{W}_1^\top, \\
\tau \frac{d\bar{P}}{dt} &= \bar{P}S^2 \left( I - \bar{P}^\top \bar{P} \right). \\
\tau \frac{d\bar{Q}}{dt} &= \bar{W}_1 \left( S - \bar{W}_2 \bar{W}_1 \right)^\top \bar{P}^\top
\end{aligned}$$

(60)

## C    LAYER IMPLEMENTATION

Our implementation and optimization were conducted entirely using PyTorch. For the RAF layers, we initialized all weights using Kaiming uniform initialization. We modified the backward pass by adjusting the gradients with respect to the input, ensuring they align with our proposed update rule. In the output layer, we learn the projection matrix $\boldsymbol{P}$ to capture the principal directions of the target labels $\boldsymbol{y}$. In the hidden layers, $\boldsymbol{P}$ is learned to project onto the principal directions of the error signal from the subsequent layer, represented as $\boldsymbol{\delta}_{l+1}$. Specifically, for a layer $l$ with input dimension $n$, output dimension $m$, and rank constraint $r$, we proceed as follows:

---

**Algorithm 1:** Modified Backward Pass for RAF Layer

---

**Input:** Error signal $\delta_{l+1} \in \mathbb{R}^{b \times m}$, activations $h_l \in \mathbb{R}^{b \times n}$, matrices $Q_l \in \mathbb{R}^{n \times r}$, $P_l \in \mathbb{R}^{r \times m}$,
      weight matrix $W_l \in \mathbb{R}^{m \times n}$

**Output:** Gradient w.r.t. input grad_input, updates $\Delta Q_l$, $\Delta P_l$, $\Delta W_l$

**Compute covariance matrix:**

**if** *use_targets* **then**
   $\lfloor$   $C \leftarrow (\boldsymbol{y} - \bar{\boldsymbol{y}})^\top (\boldsymbol{y} - \bar{\boldsymbol{y}})$;

**else**
   $\lfloor$   $C \leftarrow (\boldsymbol{\delta_{l+1}} - \bar{\boldsymbol{\delta_{l+1}}})^\top (\boldsymbol{\delta_{l+1}} - \bar{\boldsymbol{\delta_{l+1}}})$;

$\gamma \leftarrow \max\left(\mathrm{diag}\left(C\right)\right)$;

**Compute 'gradient' w.r.t. input:**

grad_input $\leftarrow \left(Q_l P_l \boldsymbol{\delta}_{l+1}^\top\right)^\top \in \mathbb{R}^{b \times n}$;

**Compute updates for $Q_l$ and $P_l$:**

$\Delta Q_l \leftarrow h_l^\top \left(P_l \boldsymbol{\delta}_{l+1}^\top\right)^\top$;

$\Delta P_l \leftarrow P_l \left(\dfrac{C}{\gamma}\right)\left(I - P_l^\top P_l\right)$;

**Compute update for $W_l$:**

$\Delta W_l \leftarrow \boldsymbol{\delta}_{l+1}^\top h_l$;

---

dRAF follows the same procedure, with the key difference being that the error signal $\boldsymbol{\delta}$ does not necessarily originate from the next layer; instead, it can come from any subsequent layer.

# D FULLY CONECTED EXPIREMENTS:

## D.1 LAYER-WISE CONSTRAINTS

In Figure [3.a], we present the results of training a network with four hidden layers, each containing 512 neurons, and an output layer with 10 neurons on the CIFAR-10 dataset. The network was trained using RAF, as described earlier, without rank constraints, except for one layer at a time. For comparison, we also trained the network using standard backpropagation as a baseline. All networks were trained with a batch size of 32, a learning rate of $6 \times 10^{-4}$, and weight decay of $4 \times 10^{-4}$. The Adam optimizer with AMSGrad was used, with training conducted for 160 epochs and an exponential learning rate decay factor of 0.975. Each experiment was repeated 10 times.

## D.2 CONSTRAINING ALL LAYERS

For the results shown in Figure [3.b], we trained the same network with four hidden layers, each containing 512 neurons. This time, we applied rank constraints to all layers simultaneously, with rank values $r = 64, 32, 16, 10$.

To further demonstrate that the network still utilizes high-rank representations, we also trained a variant of the network with 64 neurons in each hidden layer, without applying any rank constraints. All training was conducted with a batch size of 32, a learning rate of $6 \times 10^{-4}$, and weight decay of $4 \times 10^{-4}$. The Adam optimizer with AMSGrad was used, with training carried out for 160 epochs and an exponential learning rate decay factor of 0.975. Each experiment was repeated 10 times.

## D.3 CIFAR-100 SUB-SAMPLING

For the results shown in Figure [3.c], we trained the same model on the CIFAR-100 dataset, sampling different numbers of classes $d$, with $d = 50, 75, 100$. For each sub-sample, we trained the model while applying rank constraints to all layers, using various rank values. This was done to demonstrate that the dimensionality of the error signal depends on the task dimensionality $d$. All optimizations were performed with a batch size of 32, a learning rate of $6 \times 10^{-4}$, and weight decay of $4 \times 10^{-4}$. The Adam optimizer with AMSGrad was used, with training conducted for 160 epochs and an exponential learning rate decay factor of 0.975. Each training run was repeated 5 times.

### D.4 DRAF

For the results shown in Figure [3.d], we trained the same model using dRAF. In one experiment, we propagated the error signal from the last layer to all preceding layers. In a separate experiment, we propagated the error signal directly from the penultimate layer to all earlier layers, applying rank constraints to these layers while keeping the last layer at full rank. All optimizations were performed with a batch size of 32, a learning rate of $6 \times 10^{-4}$, and weight decay of $4 \times 10^{-4}$. The Adam optimizer with AMSGrad was used, training for 160 epochs with an exponential learning rate decay factor of 0.975. Each training run was repeated 5 times

## E CONVOLUTIONAL NEURAL NETWORKS

To extend our RAF algorithm to convolutional layers, we apply the rank constraint to the number of channels in the error signal. This effectively constrains the dimensionality of the error signal across the spatial dimensions. Similar to the implementation for fully connected layers in Algorithm 1, we modify the backward pass in the same way for convolutional layers.

The key difference in the convolutional context is how the projection matrix $\boldsymbol{P}$ operates on the error signals. In convolutional layers, $\boldsymbol{P}$ functions as a $1 \times 1$ convolutional filter, projecting the error signal at each spatial location from $m$ channels down to $r$ channels. This reduces the error signal's dimensionality to $r$ per pixel, adhering to the rank constraint.

As in the fully connected case, the matrix $\boldsymbol{P}$ is learned using Oja's rule. However, the covariance matrix $\boldsymbol{C}$ is computed over both the batch and spatial dimensions—that is, across all pixels in all images within the batch. This approach captures the covariance structure of pixel representations more effectively, enabling $\boldsymbol{P}$ to project the error signals appropriately in the convolutional setting.

### E.1 VGG-LIKE ARCHITECTURE

We trained a VGG-like architecture with four convolutional blocks, as detailed in Table E.2. For the results shown in Figure 4 (a), we applied a rank constraint only to the final convolutional block, which has 512 channels. For the results shown in Figure 4 (b), we constrained all layers, applying rank constraints of 1/2, 1/4, and 1/8 of the original channel dimensions. Each network was trained with the Adam optimizer, using a learning rate of $5 \times 10^{-4}$, weight decay of $5 \times 10^{-5}$, and an exponential learning rate decay factor of 0.98. We trained each model for 250 epochs, repeating each experiment five times.

### E.2 NEURAL RECEPTIVE FIELDS

For the results shown in Figure 4 d, we trained the same network as used in Lindsey et al. (2019). The network consists of two convolutional layers with ReLU activations, modeling the retina, followed by three convolutional layers with ReLU activations, modeling the ventral visual stream (VVS), and fully connected layers for classification (full architecture is detailed in Table E.2). A bottleneck is introduced between the retina and the VVS. We trained the network using Restricted Adaptive Feedback (RAF), applying rank constraints to the feedback sent to the retina with ranks $r = 2, 4, 32$. The model was trained with parameters similar to those in Lindsey et al. (2019), using the RMSProp optimizer with a learning rate of $1 \times 10^{-4}$, weight decay of $1 \times 10^{-5}$, and an exponential learning rate decay factor of 0.985. Each model was trained for 120 epochs, and all experiments were repeated 5 times.

We also conducted experiments using dynamic Restricted Adaptive Feedback (dRAF), where the feedback to the retina originated from higher visual layers while bottlenecking the feedforward weights of the retina to 4 channels, as illustrated in Figure 5. Under these conditions, we observed that the receptive fields shifted from the typical center-surround pattern expected in models with constrained forward weights to more complex patterns, as shown in Figure 5.

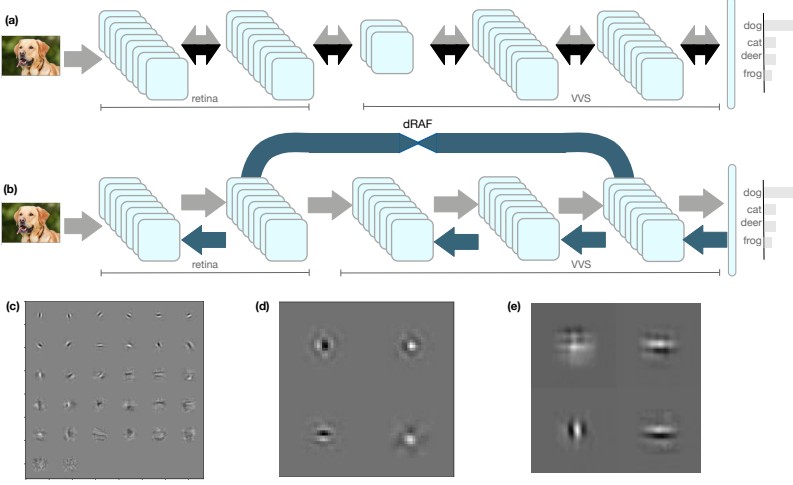

Figure 5: *Receptive Fields.* (**a, b**) Model of the early ventral visual stream, adapted from Lindsey et al. (2019), trained with BP (a) and dRAF (b). (**c**) Top receptive fields from (a) without a forward pathway bottleneck. (**d**) Top receptive fields from (a) with a forward pathway bottleneck. Constraining the retina results in more center-surround receptive fields. (**e**) Top receptive fields from (b) with a forward pathway bottleneck but without feedback constraints to the retina.

Table 1: CNN Architecture

| Layer(s) | Output Size | Details |
|---|---|---|
| Input | $32 \times 32 \times 3$ | |
| **Convolutional Block 1** | | |
| Conv2D + ReLU | $32 \times 32 \times 64$ | $3 \times 3$ conv, 64 filters, padding=1 |
| BatchNorm2D | $32 \times 32 \times 64$ | |
| Conv2D + ReLU | $32 \times 32 \times 64$ | $3 \times 3$ conv, 64 filters, padding=1 |
| BatchNorm2D | $32 \times 32 \times 64$ | |
| MaxPool2D | $16 \times 16 \times 64$ | $2 \times 2$ max pool, stride=2 |
| **Convolutional Block 2** | | |
| Conv2D + ReLU | $16 \times 16 \times 128$ | $3 \times 3$ conv, 128 filters, padding=1 |
| BatchNorm2D | $16 \times 16 \times 128$ | |
| Conv2D + ReLU | $16 \times 16 \times 128$ | $3 \times 3$ conv, 128 filters, padding=1 |
| BatchNorm2D | $16 \times 16 \times 128$ | |
| MaxPool2D | $8 \times 8 \times 128$ | $2 \times 2$ max pool, stride=2 |
| **Convolutional Block 3** | | |
| Conv2D + ReLU | $8 \times 8 \times 256$ | $3 \times 3$ conv, 256 filters, padding=1 |
| BatchNorm2D | $8 \times 8 \times 256$ | |
| Conv2D + ReLU | $8 \times 8 \times 256$ | $3 \times 3$ conv, 256 filters, padding=1 |
| BatchNorm2D | $8 \times 8 \times 256$ | |
| MaxPool2D | $4 \times 4 \times 256$ | $2 \times 2$ max pool, stride=2 |
| **Convolutional Block 4** | | |
| Conv2D + ReLU | $4 \times 4 \times 512$ | $3 \times 3$ conv, 512 filters, padding=1 |
| BatchNorm2D | $4 \times 4 \times 512$ | |
| Conv2D + ReLU | $4 \times 4 \times 512$ | $3 \times 3$ conv, 512 filters, padding=1 |
| BatchNorm2D | $4 \times 4 \times 512$ | |
| AdaptiveAvgPool2D | $1 \times 1 \times 512$ | Output size $(1, 1)$ |
| Flatten | 512 | Flatten to vector |
| **Classifier** | | |
| Fully Connected + ReLU | 256 | Linear layer, $512 \rightarrow 256$ |
| Dropout | 256 | Dropout probability $p = 0.4$ |
| Fully Connected | $C$ | Linear layer, $256 \rightarrow C$ |

Table 2: Retina model Architecture

| Layer(s) | Output Size | Details |
|---|---|---|
| Input | $32 \times 32 \times 1$ | Grayscale input |
| **Retina** | | |
| Conv2D + ReLU | $32 \times 32 \times 32$ | $9 \times 9$ conv, 32 filters, padding=4 |
| Conv2D + ReLU | $32 \times 32 \times 32$ | $9 \times 9$ conv, 32 filters, padding=4 |
| **VVS** | | |
| Conv2D + ReLU | $32 \times 32 \times 32$ | $9 \times 9$ conv, 32 filters, padding=4 |
| Conv2D + ReLU | $32 \times 32 \times 32$ | $9 \times 9$ conv, 32 filters, padding=4 |
| Conv2D + ReLU | $32 \times 32 \times 32$ | $9 \times 9$ conv, 32 filters, padding=4 |
| Flatten | $32,768$ | Flatten to vector |
| **Classifier** | | |
| Fully Connected + ReLU | $1,024$ | Linear layer, $32,768 \rightarrow 1,024$ |
| Dropout | $1,024$ | Dropout probability $p = 0.5$ |
| Fully Connected | $C$ | Linear layer, $1,024 \rightarrow C$ |

