# OpenReview forum: "Training Large Neural Networks With Low-Dimensional Error Feedback"
_ICLR.cc/2025/Conference — Submitted to ICLR 2025_

### Official Review · Reviewer_pjYe · 2024-11-01

**Soundness:** 3
**Presentation:** 3
**Contribution:** 2
**Rating:** 5
**Confidence:** 4

**Summary:**

This work explores the hypothesis that a low-dimensional error signal is sufficient to train neural networks, in particular an error signal of dimensionality equal to the neural network output, as opposed to the dimensionality of the gradient, which is equal to the number of parameters of the neural network.
It proposes a new learning algorithm that allows manipulating the dimensionality of the error signal.
The algorithm is based on feedback alignment, is local and therefore is supposed to be biologically plausible.
Training of feedback weights is not new, the novelty of the algorithm is using low-rank matrices for the feedback weights.
It studies the algorithm theoretically in a simple linear setting, it further tests it with MLP and a convnet on CIFAR10 and looks into receptive fields of neutrons.

**Strengths:**

The paper is well written and seems correct.
I find it interesting that the authors found a way to exploit the low-dimensionality of the error into a concrete algorithm, and that the algorithm can match the performance of gradient descent (backpropagation).

**Weaknesses:**

About biological plausibility, that’s a very controversial topic.
I was working in computational neuroscience for 15 years, and what I learned is that there is absolutely no consensus on which model is biologically plausible which one is not.
The word “”biologically plausible” is used by different people with vastly different meanings, so much so that those words are meaningless unless used in a very restricted context of a few “friends” who work on the same identical problems.
Therefore, in my opinion, the “biological plausibility” adds very little value to this paper.

About the machine learning side: the fact that the error signal should have dimensionality equal to the neural network output is trivial, and can be studied in any neural network, not just in the linear setting.
Using simple chain rule, it is easy to find that the gradient of the loss is equal to J^T err, where J is the Jacobian matrix of the neural network with respect to the parameters, and err is the vector of errors (the gradient of the loss with respect to the output).
The vector err has dimension d, where d is the dimensionality of the output, while the matrix J has dimension d x p, where p is the number of parameters.
Therefore, for any neural network, it is obvious that the gradient is a projection of a low dimensional (d) vector into a high dimensional space (p).

Trivial or not, I find it interesting that the authors found a way to exploit the low-dimensionality of the error into a concrete algorithm.
However, does this algorithm represents an advance for machine learning?
Feedback alignment is not new, training feedback weights is also not new, so the only novelty seems to be the low rank of the feedback weights.
That does not seem to be a significant contribution.

**Questions:**

I wonder whether the low-dimensionality of the error signal may be exploited to design an algorithm that is actually useful.
The proposed algorithm may perhaps save a little bit of compute during the reverse pass, due to the low-rank of the feedback weights, but it requires more memory, because it requires more weights than a standard neural network (feedback weights in addition to forward weights) and it still requires saving neural activations during the forward pass like standard models.

---

> ### Author Response · Authors · 2024-11-19
>
> Thank you for your feedback. We are encouraged that you find the work well-written and see the potential of our ideas and theory.
>
> 1. We couldn’t agree more with you about the overuse (and perhaps misuse) of the term “biological plausibility”.  It is not well-defined, unclear, and, more importantly, untrue. Most models that claim biological plausibility probably cannot be implemented by real neurons without significant adjustment and extensions (and we include our model here). However, the term is widely used and has effectively become a jargon. Authors often use relative terms (i.e., “more biologically plausible”), which is also inaccurate.
> Nevertheless, **we are happy to replace that phrasing with a more detailed description**. In particular, we have removed unnecessary uses of “biological plausibility” and replaced it with what we actually mean: we study how large neural networks learn using indirect restricted error signals. In the brain, a neural population is unlikely to receive a detailed error signal that originates from an efferent source. Thus, it is crucial to understand how error constraints affect learning and shape neural representations.
>
> 2. Regarding your second point, from the machine learning perspective. We believe there is a misunderstanding. You are correct that in linear networks, the error dimensionality increases and is equal to the dimensionality of the output error. In this case, it is unsurprising that we can train with low-rank feedback because the error dimensionality does not change. However, we still gain a lot of insight from the linear networks. Particularly, we see that low-rank feedback matrices must learn or “align” to the error space. This is because they must send the correct error to afferent layers.
> Developing your idea, studying the dimensionality in the deeper layers of a nonlinear network is potentially possible, assuming random connectivity and using more sophisticated techniques of dynamic mean-field theory or the dynamic cavity method [Clark, Abbott, and Litwin-Kumar. 2023]. This, however, would be the subject of future work.
>
> However, in nonlinear networks, the dimensionality of the error varies between the layers and during training. **The Jacobian depends on the activity in every step, effectively increasing the error subspace.**
> Surprisingly, the learning rules we develop using our intuition from linear networks work in the nonlinear case. We believe it is because our basic logic is correct. **We’ve elaborated and clarified this point in Section 2.**
>
> 3. Finally, we emphasize that we do not claim nor did we attempt to derive computationally efficient algorithms or state-of-the-art performance. Instead, we show that large networks can be trained using indirect and restricted error signals if (and only if) the feedback matrix learns (i.e., “aligns” also the efferent error space). Nevertheless, we point out that training with low-rank feedback is not significantly less efficient than BP (unlike other methods of FA). Consider training the weights between two large layers with $M,N\gg1$ Neurons each. This requires learning $O(MN)$ parameters. The low-rank feedback matrix requires learning additional $O(r(M+N))$ parameters, which is negligible in large networks (assuming $r$ is small).  Thus, while we do not make any efficiency claims, we can assert that it has not significantly increased compared to BP and would be smaller than other FA-based methods.
>
> **References**
> - Clark, D. G., Abbott, L. F., & Litwin-Kumar, A. (2023). Dimension of activity in random neural networks. *Physical Review Letters*, *131*(11), 118401.

---

> > ### Comment · Reviewer_pjYe · 2024-11-22
> >
> > Thank you for pointing out that the Jacobian changes during training in nonlinear models, that is of course correct. However, there are some known circumstances in which the projection of the error would stay in a low dimensional space in nonlinear models, for example in neural networks with large width. That has been extensively studied using the neural tangent kernel, in particular when it remains approximately constant during training. It may be that the results of your experiment hold only when neural networks are overparameterized and are somewhat trivial given what we know already on the large bulk of literature on NTK. Can you please comment on that?

---

> > > ### Author Response · Authors · 2024-11-22
> > >
> > > Thank you for your response. You raise a good and valid question since NTK relies on linearizing deep neural networks in the lazy regime. However, there are several arguments for why error dimensionality cannot be explained by NTK theory:
> > >
> > > 1. NTK theory approximates large neural networks as **linear in their parameters** around the initialization point. However, parameter space linearization does not directly imply linearity in the response. Since the response enters the jacobian we discussed above, NTK does not explain linear error backpropagation.
> > >
> > >
> > > 2. NTK theory considers the divergence across all parameters, i.e., all weights in all layers. To the best of our knowledge, it has not been applied to study changes from layer to layer. **Layerwise analysis is crucial** for describing the dimensionality of the error as it backpropagates through the network.
> > >
> > >
> > > 3. Our experiments show that low-dimensional error signals efficiently train networks with layers as small as 64 neurons (or channels in CNNs)—**likely outside the scope of NTK theory.**
> > >
> > > That said, your comment leads to an interesting proposition: are there conditions in the mean-field limit under which we can linearize the response for small errors? One possibility is to consider the limit in which the Jacobian eigenvalues are small. While an interesting idea for further study, it is beyond the scope of this paper.
> > >
> > > Finally, we note that even i**f we can approximate a large nonlinear network as linear in the error dynamics, it will only strengthen our results.** Such an approximation would mean that our exact analysis of linear networks can be applied, under some conditions, to nonlinear networks.

---

> > > > ### Author Response · Authors · 2024-11-25
> > > >
> > > > We hope we have addressed your questions and eased most of your concerns. Please let us know if you still have questions we could answer in the remaining discussion time. Thank you.

---

> > > > > ### Comment · Reviewer_pjYe · 2024-11-26
> > > > >
> > > > > Let's refer to a specific paper for concreteness, out of many papers published on the topic of NTK.
> > > > > Let's take "Wide Neural Networks of Any Depth Evolve as Linear Models Under Gradient Descent" by Lee et al 2019.
> > > > > Equation 2 is true in general, including nonlinear models, and is basically the equation I described in my review above.
> > > > > Equation 6 is the same, but now the Jacobian does not change during training, because training is lazy, parameters stay close to their initial values.
> > > > > It is important to note that the Jacobian has dimension nd x p (n: number of data points, d: output dimension, p: number of parameters).
> > > > > There is a difference with respect to what I wrote in my review above, there I wrote the Jacobian as a d x p matrix, but that is valid only for the case of one data point (n=1).
> > > > > Now, in the lazy regime, the error lives in a constant subspace of dimension nd, that may be much smaller than p when the model is overparameterized.
> > > > > This subspace does not change during training, however it has dimension nd, instead your work suggests that we can get this dimension down to d.
> > > > > In truly linear models, the dimension is d, instead of nd, because the Jacobian does not depend on data, therefore different data points correspond to the the same subspace of dimension d.
> > > > > Using this framework, in the nonlinear case, can you please help me to understand how the error dimension is equal to d instead of nd? Do different data points correspond to the same subspace of dimension d even if the model is nonlinear?
> > > > >
> > > > > If you can answer this question, then I would be a step closer to understand your results even in the underparameterised case (since you state that low dimensional error holds in your experiments for underparameterized models as well).
> > > > > In that case, training is not lazy and Jacobian changes during training.
> > > > > However, if I understand why the subspace of dimension nd is reduced to dimension d, even if this subspace changes during (non-lazy) training, then I can imagine that your low-rank feedback weights would be able to track changes in this subspace, since those weights are trained and thus also change during training.

---

> > > > > > ### Author Response · Authors · 2024-11-27
> > > > > >
> > > > > > Thank you for engaging in this discussion. We find it quite clarifying—and we hope you do, too.
> > > > > >
> > > > > > - We agree with your analysis of the NTK equations from [Lee et al. 2019].  **We do not claim or think that backpropagation in non-linear networks can be approximated as low rank** (although a recent study found that backpropagation is biased towards lower-rank solutions [Patel and Swartz-Ziv, 2024]).  An exact solution for the dimensionality in nonlinear networks could be obtained using dynamic mean-field theory ([Clark et al., 2023]), done in the lazy regime.
> > > > > > - To be clear, we are not claiming that the nonlinear networks are approximated as linear. Rather, we first develop a theoretical framework for linear networks and then test whether our insights apply to more complex networks (nonlinear, convolutional, direct feedback). We find that **our learning rules, derived from linear analysis, can successfully train nonlinear networks**. This methodology has been used in many theoretical studies (e.g., [Saxe et al. 2015, Li and Sompolinsky 2021, Refinetti et al. 2021]). We assert that it is because our central insights from the linear analysis hold in nonlinear settings.
> > > > > > - What are the insights that carry to nonlinear networks? First, we note that in a shallow linear architecture with quadratic error, the Johnson–Lindenstrauss lemma implies that any random feedback matrix with rank $r>d\log M$ would be sufficient for training, where $M$ is the size of the penultimate layer and $d$ is the output error dimensions. This is because, according to the JS lemma,  the bottleneck in the feedback maintains correlation and the error structure.  However, our dynamical analysis shows that *this is not the case, even in linear systems*. The problem is that with low-rank feedback, the learning dynamics (which is nonlinear [Saxe et al. 2015]) have multiple suboptimal solutions. This observation leads to the first important result of our work: when the error pathway is constrained, it needs to align itself  (the row space of the feedback matrix) with the error subspace at the output. This result is at the heart of our learning rule, and we claim that it extends to nonlinear networks.
> > > > > > - We then tested our learning rule numerically on nonlinear networks and showed it is sufficient to match backpropagation. **The conclusion is NOT that overparameterized nonlinear networks can be approximated by a linear network, but rather that low-rank feedback—when properly aligned—is sufficient to train them.** As our linear analysis shows, it is not trivial. Furthermore, we show that aligning the feedback can be done locally, for example, by using Oja’s learning rule, which learns the top components of the error.
> > > > > > - How exactly the low-rank matrix trains nonlinear networks remains to be studied. Your explanation that the feedback matrix tracks the error is possible, though we do not believe it changes fast enough to adjust with each sample. Instead, we have a different intuition that follows the intuition of Feedback Alignment. In over-parameterized networks, even the rough low-dimensional estimation of the error can push the network to the highly degenerate minimum loss. The crucial point is that the low-rank feedback needs to align with the error signal in the efferent layer, or it will learn suboptimally (as in the linear networks).
> > > > > >
> > > > > > We hope this summary clarifies our results and claims regarding the nonlinear networks and emphasizes the novel insights of our work.

---

> > > > > > > ### Author Response · Authors · 2024-12-03
> > > > > > > **Final words for our discussion**
> > > > > > >
> > > > > > > Thank you for actively engaging in the discussion—your thoughtful feedback has been invaluable. In a year where such in-depth exchanges with reviewers seem disappointingly rare, we deeply value the opportunity to clarify and improve our work. We hope our response has addressed your last concern, particularly regarding why the ability to train nonlinear networks with low-rank feedback is not due to effective linearization of error dynamics and why the extension to nonlinear networks is nontrivial and significant.
> > > > > > >
> > > > > > > We trust that, through our exchange and the revisions we’ve made, we have demonstrated that our results are solid and noteworthy. Should you feel that our responses and updates have sufficiently addressed your concerns, we would greatly appreciate any reconsideration of your score.
> > > > > > >
> > > > > > > Thank you,
> > > > > > > The anonymous authors.

---

### Official Review · Reviewer_KBDT · 2024-11-04

**Soundness:** 3
**Presentation:** 2
**Contribution:** 2
**Rating:** 8
**Confidence:** 4

**Summary:**

The paper aims at approximating backpropagation by using a low-rank projection of the forward matrix (and hence an approximation of the true gradient). The backward pass in each layer is done via two layers that together form a low-rank approximation of the forward weights. These weights are aligned to the forward weights through a combination of Hebbian and Oja’s rules.

**Strengths:**

1. The proposed method performs well and also provides a good approximation to backprop.
2. Sec. 5 results are interesting, as they suggest the same type of phenomena can be due to either forward or backward architectural choices (but see below).
3. Overall, this result shows that error signals have to at least match the task dimensionality for efficient learning, which provides useful intuition for bioplausible learning search.

**Weaknesses:**

1. Small improvement compared to [Akrout 2019]

The proposed approach doesn’t add much to the discussion in [Akrout 2019], at least in my opinion. [Akrout 2019] (using Eq. 10 (top) for $P=I$ in the definitions of this paper) showed that the Kolen-Pollack rule is good enough to achieve backprop-level performance on ImageNet. The proposed approach introduces one more feedback layer per forward layer, which is arguably even less biologically plausible than the feedback alignment-style feedback network.

2. Oja’s rule issue

As presented in the paper, the learning rule for $P$ is not local – it requires weight transport, exactly the issue the authors are solving for backprop. This is because the matrix version of Oja’s rule uses a $(P\delta)(P\delta)^T P$ term: $P\delta$ is the output of the layer receiving $\delta$, so, like in backprop, $(P\delta)(P\delta)^T P$ requires the output activations to be multiplied by the transpose of the weights to find $(P\delta)^T P$. It might be that you can still do PCA with some modifications of Oja’s rule that make it local, but that should be clearly articulated in the paper. (What’s usually considered local is the 1d version of Oja’s rule, since it doesn’t have matrix multiplication. But that can only find the top principal component.)

3. No baselines for Sec. 5

Sec. 5 is an interesting approach, but it doesn’t have baselines. The authors should show that the baseline models (i.e. backprop, same architecture) with/without feedforward bottleneck produce different receptive fields. I understand that the architecture/training is the same or similar to [Lindsey 2019], but a reader shouldn’t have to look at another paper to find the baselines. (It is especially important since the results is purely qualitative)


4. Some parts of the literature are misrepresented

Lines 91-98 first introduce [Akrout 2019], and then state that it “struggles to match BP performance in complex architectures like CNNs”. Fig. 3 in [Akrout 2019] shows that their approach works exactly as well as backprop on ResNet50 trained on ImageNet.

Lines 416-420 mention that training CNNs with feedback alignment is hard, but that adaptive feedback “has made progress”. See above regarding [Akrout 2019], but even earlier work like [Liao 2016] (the sign-symmetry method) has achieved significant progress in that direction. Both should be cited in this paragraph.

Finally, line 491 says “our novel learning rule bears superficial similarity to previous Feedback Alignment (FA) schemes”. I don’t think it’s fair to call Eq. 10 only superficially similar to Kolen-Pollack, as it is pretty much the same rule with one extra step.

**Questions:**

1. Is there any way to quantify Sec. 5 results? The shift between center-surround and orientation selective is interesting, but presumably it is a function of how narrow the bottleneck is, and also of how wide the network is. Having a quantitative plot for these results would be a good addition to the paper.
2. In Eq. 10 (left), shouldn’t $h_l$ be $h_{l-1}$?
3. Small issues: figures should have larger font sizes as they’re hard to read; Fig. 5 in the appendix has a placeholder caption; typo in "necessary" in line 215

-------------
Updated the score to 8; see the rebuttal discussion.

---

> ### Author Response · Authors · 2024-11-19
>
> Thank you for taking the time to read and thoroughly comment on our work. The strengths you mention increase the potential benefits of our applied results. We believe our work also holds novel theoretical insights, and we hope we can convey them with our responses to your comments and questions
>
> **Response to the mentioned weaknesses:**
> 1. We emphasize that **we address a different problem than [Akrout 2019]**. In our model, the low-rank feedback matrix constrains the feedback error dimensionality. This is central to our model since we focus on how networks can learn with a low-dimensional, restricted error signal. Setting $P=I$ is equivalent to [Akrout 2019] is the rank is full (i.e., the rank of $B=QP$ is $\min(N,M)$. Furthermore, we show that if the matrix is low-rank. This learning rule will not work. This observation leads to the key difference from [Akrout 2019] and other previous models: when the error pathway is restricted, the feedback matrix must also learn its input space. In other words, the feedback should not only align with the feedforward weights (as in [Akrout 2019], [Lilicrap 2016], and others) but also align with the error. Our algorithm aligns it both with the weights and the error.
> In addition, we are not sure how to gauge if adding an additional feedback layer is “less biologically plausible”. We note that this feedback layer will only have $O(d)\ll N$ neurons. Several previous studies has suggested adding interneurons as a mean to implement gradient descent (e.g., Leinweber et al. 2017 or Sacramento et al. 2018)
> 2. This is a correct and important observation. The learning rule relies on the transposition of the $P$ matrix, the same way that Oja’s rule requires it. Implementation of this rule, as defined in eq. (8) violates synaptic locality due to the $P^TP$ term. Failing to highlight it was an oversight. However, we argue that this point does not reduce the significance of our results:
>     1. First, we do not claim we solved the problem of weight transfer. We note that BP suffers from weight transfer, which was the original motivation behind FA. **We now clarify this point in the text**.
>     2. While our update rule involves matrix transposition, it does not require a backward pass like backpropagation, where the signal is propagated backward through the axons. Likewise, it does not require keeping an exact copy of the transposed weights to propagate the error backward.
>     3. **Possible biological implementations and approximations for Oja’s rule exist.** The biological plausibility of the learning rule has been discussed in original work by [Oja 1982], but also more modern variants, including homeostatic plasticity mechanisms [Clopath et al. 2010], synaptic rescaling [Turrigiano 2008], or similarity matching [Pehlevan and Chklovskii, 2015].
>     4. Most importantly, the biological plausibility of the learning rule **does not change our work's central messages and conclusions**. In particular, we show that large neural networks can learn with minimal, restricted error signals if the feedback matrices align with the error space. In that case, we show that it matches BP’s performance. Finally, we show that the error signal dimensionality shapes the receptive fields.
>
> 3. Point taken. **We have added the results of [Lindsey 2019]** in our manuscript for reference.
>
> 4. Following your advice, **we have clarified and elaborated on how previous work relates to this study**. In particular, we point out that [Akrout 2019] and [Liao 2016] have made significant advances, but they have not addressed the problem of restricted error signals. Section 3 now explains that these methods are **insufficient for training with a low-rank feedback matrix**.
>     Finally, while our learning rule composes only one extra step from the Kolen-Pollack framework, we believe there is a conceptual advancement. Previous studies use plasticity rules to align the feedback matrix with the feedforward weights. Here, we claim that if the error channel is restricted (as in the case of low-rank feedback),  the weights must align the row space with the error space. While this is achieved using a single step, it represents an important theoretical idea of what the feedback channel does and is a departure from the Kolen-Pollack framework. Nevertheless, we have removed the word ‘superficially’ from the text.

---

> ### Author Response · Authors · 2024-11-19
>
> **Answer to specific questions:**
>
> 1. This is a good question. Quantifying how “center-surround” the receptive fields are is challenging. Since the receptive fields are noisy, quantitative analysis requires training many networks, which is beyond our current capacity due to time limitations. We note that [Lindsey 2019] also did not quantify this observation. Importantly, we emphasize that we did the expected control: **we trained a network with a bottleneck in the feedforward while allowing full-rank error propagation** (i.e., we used our training but did not restrict the rank). In this case, we saw no center-surround receptive field. These fields emerged when and only when the error was constrained. These results are reported in the appendix.
> 2. Fixed.
> 3. Fixed.
>
> We hope that we have conveyed the theoretical significance of our work and its conceptual advance over previous studies of FA methods. As soon as we integrate all comments and proofread, we will upload the corrected PDF.

---

> ### Author Response · Authors · 2024-11-19
>
> **References**
>
> - Leinweber, M., Ward, D. R., Sobczak, J. M., Attinger, A., & Keller, G. B. (2017). A sensorimotor circuit in mouse cortex for visual flow predictions. *Neuron*, *95*(6), 1420-1432. (and the relevant erratum)
> - Sacramento, J., Ponte Costa, R., Bengio, Y., & Senn, W. (2018). Dendritic cortical microcircuits approximate the backpropagation algorithm. *Advances in neural information processing systems*, *31*.
> - Clopath, C., Büsing, L., Vasilaki, E., & Gerstner, W. (2010). Connectivity reflects coding: a model of voltage-based STDP with homeostasis. *Nature neuroscience*, *13*(3), 344-352.
> - Turrigiano, G. G. (2008). The self-tuning neuron: synaptic scaling of excitatory synapses. *Cell*, *135*(3), 422-435.
> - Pehlevan, C., Hu, T., & Chklovskii, D. B. (2015). A hebbian/anti-hebbian neural network for linear subspace learning: A derivation from multidimensional scaling of streaming data. *Neural computation*, *27*(7), 1461-1495.
> - Liao, R., Schwing, A., Zemel, R., & Urtasun, R. (2016). Learning deep parsimonious representations. *Advances in neural information processing systems*, *29*

---

> ### Comment · Reviewer_KBDT · 2024-11-19
> **Comments on the rebuttal; upping score to 8**
>
> Thank you for a detailed response!
>
> 1. Contribution compared to [Akrout, 2019]
>
> I agree that low-dimensional feedback is not the topic that paper was solving, although I still think that conceptually the most important part of the FA-insipred line of work is the basic Kolen-Pollack algorithm.
> > In addition, we are not sure how to gauge if adding an additional feedback layer is “less biologically plausible”.
>
> Here's my intuition: for a basic FA-style algorithm, you need a separate error network that mimics all connections in the forward pass. I don't think we have evidence for such connectivity in the cortex. In your case, that network has more hidden layers/neurons (although fewer weights) to implement the same algorithm (backprop), while still requiring one-to-one mapping between forward and backward neurons.
>
> That said, I think your work simultaneously relaxes connectivity constraints by allowing less direct error pathways, which is interesting conceptually.
>
> 2. Oja's rule
>
> Thanks for the references. I now recall the [Pehlevan and Chklovskii, 2015] solution, and explaining Oja's rule implementation(s) would definitely strengthen the paper.
>
> However, I disagree that your work is not trying to solve the weight transport problem -- weight transport is exactly why you need to approximate the forward weights. I think your low-rank discussion is a consequence of trying to solve the weight transport problem. (This doesn't make your point weaker, in my opinion, but I think this view better captures your contribution.)
>
> Also, thank you for addressing the small issues/presentation concerns I had.
>
> Overall, I think this work is interesting and contributes to the discussion on potential learning mechanisms in the brain. I'm raising the score from 5 to 8 as most of my concerns have been addressed.

---

### Official Review · Reviewer_QAwR · 2024-11-04

**Soundness:** 2
**Presentation:** 1
**Contribution:** 2
**Rating:** 3
**Confidence:** 5

**Summary:**

This paper introduces Restricted Adaptive Feedback (RAF), a learning method inspired by Feedback Alignment (FA) using low-dimensional error feedback to train neural networks.
By adopting a factorized version of Feedback Alignment with low-rank matrices, the need for high-dimensional error signals is challenged.
RAF achieves performance comparable to backpropagation in some settings, offering a low-dimensional biologically plausible training principle.

**Strengths:**

1. The subject of biological alternatives to back-propagation (BP) is a really interesting topic.

2. The authors provide good theoretical foundation of their method.

3. The study shows that error dimensionality shapes receptive fields which is interesting and provides insights on the emergence of representations in both artificial and biological systems.

4. The empirical study shows competitive performance with reduced error signal dimensionality, which could reduce computational costs without sacrificing accuracy.

5. The link between task dimensionality and the minimal rank is interesting and promising.

**Weaknesses:**

1. The paper is not very well-written in multiple aspect. The use of figurative language and lack of precision makes it somewhat ambiguous and less rigorous and I would consider rephrasing to focus on the main concept in a more direct way.
This can be seen in the first paragraph of the introduction for example. The paper is sometimes very redundant, making the read feel repetitive and unnecessarily lengthy. The notations are repeated (eg. numerous mentions of $\mathbf{x}$ being the input vector).
A thorough proofreading is needed (example Figure 5 in the Appendix, the caption is "Enter Caption").

2. The study is extensive on easy/non practical settings (linear/shallow cases) which leaves limited room for presenting interesting and novel results in more complex scenarios.

3. Several references are lacking, which is very detrimental as some previous experimental results not presented have adapted FA/DFA to a wide range of architectures. For example (Akrout et al., 2019) is merely cited but they present adaptations of FA that match back-propagation on more complex architecture (ResNet-50) and on more complex data (Image-Net) than the presented results. Furthermore, (Sanfiz and Akrout, 2021) which benchmarked FA, DFA and other biologically plausible alternatives to BP with open-sourced code is not cited.
Another work by (Launay et al., 2020), scaling to some degree DFA to even more complex architectures such as Transformer is not mentioned either.
Altogether, this makes the experimental part very shallow, which is unfortunate because the undelying idea has a really good potential and the consequence of the main difference with the other adaptations (sparser feedback matrices -> lower memory constraints) is not studied.

**Questions:**

1. You mention the use of batchnorm in your convolutional network (Table 1 in the Appendix). How could it be interpreted biologically? How do you train the parameters of the batchnorm in your model? Do you use RAF? Do you set the affine to false?
 How could maxpooling be biologically interpreted? Same question with dropout as you use a probability of 0.4.

2. Could your findings be linked to the fact that you only tested with image data? Could the dimensionality reduction be transposable to natural language processing (NLP)? Would the task dimensionality determine the minimal rank ? How could it then be adapted to learn in a self-supervised manner?

---

> ### Author Response · Authors · 2024-11-19
>
> Thank you for reading and thinking about our work; we appreciate your input. We are happy you found the strengths of our work as we see them. We believe that these points, as you summarize them, serve as a significant step towards a better understanding of learning in brain circuits.
>
> We are sorry that you find the paper not very well written. We aim our text for a diverse audience from different fields interested in learning representation, specifically because of our work's strong biological motivation and implications. We are aware of the tradeoff that by doing so
>
>
>
> **Response to the mentioned weaknesses**
> 1. We are sorry to hear that you find the manuscript not well written. Our goal was to make it accessible to a wide spectrum of readership. This was particularly important to us as we focus on biological implications, which may interest readers from various fields and levels. We are aware of the tradeoff that advanced readers may find our language repetitive and perhaps too figurative. Nevertheless, it is intended to be correct and full. We appreciate your specific comments. The caption on Fig. 5 was a mistake we corrected. The repetitive definitions (as for the input $x$) are intended to guide a less familiar audience.
> 2. You are correct that the paper emphasizes the linear theory. It allows a full solution to the learning dynamics. **Crucially, it reveals why the learning fails when the rank is low: the learning dynamics have multiple fixed points, which are not the correct solution. It also helps us identify the learning rule that prevents it—the feedback weights must also "align “ in the error space**. Our work is mainly theoretical and explains how neural networks can learn with very limited error signals, which we expect to be the case in the brain. The intent was not to match SoTA performance on complex data sets. We also note that many seminal theoretical papers rely heavily on linear theory to explain the dynamics and behavior of nonlinear networks (e.g., Saxe et al. 2013, Refinetti et al. 2021, or Li and Sompolinsky 2021).
> 3. We thank you for highlighting these references, which we did not include. However, **none of these previous studies have addressed the problem of restricted (low-rank) error pathways**, which is important for computational neuroscience. Particularly, we point out that Akrout et al. did not address this problem. Furthermore, we show our manuscript that **Akrout and other FA solutions fail with a restricted error pathway** as they all lack a crucial ingredient of “aligning” to the error space.
>
> **References**
> - Saxe, A. M., McClelland, J. L., & Ganguli, S. (2013). Exact solutions to the nonlinear dynamics of learning in deep linear neural networks. *arXiv preprint arXiv:1312.6120*.
> - Refinetti, M., Ingrosso, A., & Goldt, S. (2023). Neural networks trained with SGD learn distributions of increasing complexity. *International Conference on Machine Learning*
> - Li, Q., & Sompolinsky, H. (2021). Statistical mechanics of deep linear neural networks: The backpropagating kernel renormalization. *Physical Review X*

---

> > ### Comment · Reviewer_QAwR · 2024-11-22
> >
> > Thank you for your comment on the weaknesses I pointed out. I am convinced the paper has potential as the idea of low-rank error feedbacks is really promising. I am however not convinced yet by your answers and am happy with my grade for now.
> >
> > 1. I am convinced the paper has potential, but the current form is still lacking.
> > One thing that is striking in the paper (and your rebuttal) is the repetitive mention of **"biological plausibility/motivation"**, which is overall debatable as there is no real meaning to this except in very specific settings (and the corresponding meaning can vary). Although the original paper of Feedback Alignment is motivated by it, I find it somehow a bit too redundant if not more backed up by neuroscience works.
> > Altogether with vague and figurative sentences (as pointed out in original weakness 1) somehow frames the paper more like a position paper than a theoretical one.
> >
> > 2. It is a fair point you mention, however in their theoretical analysis, (Refinetti et al. 2021) use the sigmoidal activation function. I am not convinced that an extensive study of Deep Linear Networks should take a **major part** of the main paper here. An overview of the results and a tentative to extend those results to non-linear networks (even under some assumptions) would be more convincing.
> >
> > 3. It is true that **(low-rank) error pathways** is a really interesting idea and strength of the paper and positions it favorably with previous works I mentioned.
> > However the experimental part is still a bit too shallow. I am not looking for state of the art architectures of course, but more advanced architectures like resnets (even small ones). Furthermore, no study of the computation gains of the method (even just theoretical) is provided.
> > Lastly, and this is linked to my first point, as you mention biological plausibility throughout the paper, some could wonder why you do not study spiking neural networks that are *supposedly* much closer to how the brain works than standard ones.

---

> ### Author Response · Authors · 2024-11-19
>
> **Answers to specific questions**
>
> 1. These are valid concerns since none of these methods has variable biological implementations. First, we emphasize that the fully connected nonlinear networks did not use Maxpooling, batchnorm, or dropouts. However, these techniques stabilized and improved training in the full convolutional network. Note that the same is used for the BP we compare. Without these regularization methods, both RAF and BP show poorer but still similar performance. We emphasize that we did not claim or attempt to solve how the brain learns or implements backpropagation. This work focuses on a specific question: **can large networks learn using a low-dimensional error signal, which is more likely to be implemented in the brain?**
> 2. Our results state that we can restrict the error dimensionality when the error dimensionality of the output is low. While it sounds trivial, it is not since the credit assignment using backpropagation implies the error in each layer is high-dimensional. **We specifically used data sets with low-dimensional readout errors.** We also specifically looked at the effects of the output error dimensionality by subsampling the CIFAR-100 dataset. We do not know if our result will extend to LLMs since these produce high-dimensional error signals. If they do, it is not due to the learning dynamics but to degeneracies in the error landscape of these models, which were not the focus of this work.
>
> We have added the references you highlighted in our introduction and discussion and clarified the scope of our work. We are integrating these changes with other comments and will upload a revised version shortly.

---

> ### Comment · Reviewer_QAwR · 2024-11-22
>
> Thank you for tentatively addressing the questions I asked.
>
> 1. The answer is not really convincing. My question was aimed to see if you had intuitions about how some operations well-suited for artificial neural networks could have similar counterparts in the learning of the brain. Furthermore, although it may seem self-evident, why does a "low-dimensional error signal is more likely to be implemented in the brain"? Is this a claim or do you have references that study this interesting result?
>
> 2. Thank you for your answer. I thought your method could be applied to (relatively) small language models when fine-tuned on classification tasks. This study could balance the experimental part and maybe show that your results stand for image data and not language data, highlighting that different data-dependent learning dynamics exist. That would provide a great insight, though I understand it might not be feasible in the short amount of time you have during this phase.

---

> > ### Author Response · Authors · 2024-11-27
> >
> > Thank you for reading our response and clarifying your concerns.
> >
> > 1. **Importance of linear analysis**. As demonstrated (Refinetti et al. 2021, Saxe 2015, Li and Sompolinsky 2021), in which the *theoretical* analysis focuses on the linear network, we believe that novel insights can be gained from linear networks. Why is the linear case interesting? First, we can derive a full mathematical theory. Second, in a shallow linear architecture with quadratic error, the Johnson–Lindenstrauss lemma implies that any random feedback matrix with rank $r>d\log M$ would be sufficient for training, where $M$ is the size of the penultimate layer and $d$ is the output error dimensions. This is because, according to the JS lemma,  the bottleneck in the feedback maintains correlation and the error structure.  However, our dynamical analysis shows that *this is not the case, even in linear systems*. The problem is that with low-rank feedback, the learning dynamics (which is nonlinear [Saxe et al. 2015]) have multiple suboptimal solutions. This observation leads to the first important result of our work: when the error pathway is constrained, it needs to align itself  (the row space of the feedback matrix) with the error subspace at the output. This result is at the heart of our learning rule.
> > 2. **Complexity of biological models.** We agree that it would be interesting to see if our ideas extend to even detailed models of learning in biology, e.g., networks of spiking neurons. **However, we don’t think that we need to implement spiking neural networks to advance our understanding of learning in the brain.** Many challenges in training spiking neurons are unrelated to the error feedback they receive. The complexity of biological systems across all scales. **We believe we chose an appropriate scope for the problem we tackle.**
> > 3. Finally, we agree that “biologically plausible” is often overused and misused terminology (see our response to the 4th reviewer below). As a result, we have replaced most of the references to “biologically plausible” with what we mean: we study how large neural networks learn using indirect restricted error signals. Most error signals in the brain originate from subcortical areas and deep nuclei through long-range connections that are typically narrow (e.g., Inferior Olive, Substantia Nigra, VTA, Cerebellar deep nuclei, Anterior Cingulate Cortex). **Thus, it is crucial to understand how error constraints affect learning and shape neural representations.** Our model, while far simpler than brain circuits, is the simplest model to study the effects of constrained error signals.

---

> > > ### Comment · Reviewer_QAwR · 2024-12-02
> > >
> > > Thank you for your answers and for taking into account some of my concerns.
> > > I will however keep my rating as it is for now as I think the paper is not good enough yet and would benefit from stronger contributions in a future submission.
> > >
> > > 1.  some claims about related works seem false.
> > > > l.97/98: "better aligning forward and backward weights, it still requires high-dimensional error signals and
> > > struggles to match BP performance in complex architectures like CNNs."
> > >
> > > The first part of the sentence is correct even though an interesting but probably highly complex ablation study would be to try the method proposed by (Akrout et al.,  2019) in a low-dimensional error signal context.
> > > The second part however is simply false as the mentioned paper reports results equivalent to back-propagation when training a ResNet-18 and a ResNet-50 on ImageNet.
> > >
> > > 2. Your theoretical insight is interesting but should be extended
> > >
> > > If I am not mistaken, the Johnson-Lindenstrauss (JL) lemma provides a sufficient condition for **static embedding properties** but does not imply sufficiency for dynamic feedback processes in training neural networks. In fact what your analysis show is that in the context of dynamic feedback process, this is not the case.
> > > If I remember correctly the paper by (Refinetti et al., 2023), their analysis is on non linear neural network, using a Taylor expansion of the activation function to help their analysis. The same be done in your case to open up the discussion on non linear neural networks.
> > > Further interesting work came to my mind when reading your previous comment [1] that might also help extend your analysis to non linear convolutions (I am completely unrelated with the authors, but using their work could help you have insightful results).
> > >
> > > 3. The experimental part is too shallow
> > >
> > > I really encourage the authors to expand their experimental framework to more architectures (ResNets as done by (Akrout et al.) for example), but also to other "field".
> > > Classification exists in Natural Language Processing (NLP) and it would be really convincing to get an analysis on this kind of task.
> > >
> > >
> > >
> > > [1] A Johnson-Lindenstrauss Framework for Randomly Initialized CNNs, Nachum et al., 2022

---

### Official Review · Reviewer_afy6 · 2024-11-04

**Soundness:** 3
**Presentation:** 3
**Contribution:** 3
**Rating:** 6
**Confidence:** 5

**Summary:**

This manuscript focuses on developing a feedback alignment method that uses low-rank gradient approximation matrix, B, in order, to show that low-dimensionality error is sufficient for learning.

**Strengths:**

- Theoretical extensions of the Kolen-Pollack algorithm are very clear and easy to follow. Theoretical contributions were validated well on simulations in linear models.
- Experiments on CIFAR-10 with different numbers of classes seem interesting given the dimensionality match between the number of classes and the rank of the matrix B.
- The experiments on the receptive fields show an interesting relationship between features learned by the model and the rank of the gradient.

**Weaknesses:**

- Given the difficulty of FA to match higher class datasets (as shown by Lillicrap et al), the paper would benefit by doing experiments in ImageNet as in Akrout et al, 2019. Similarly to the CIFAR100 experiments, the rank of the B follows the number of categories in ImageNet. Would it do it on other datasets such as SVHN?
- Novelty of low-rank gradients. There has been extensive work on the implicit regularizarion literature (see examples below), that show that gradient descent is regularized to change the weights of the model in a low dimensional space:
   - Ma, C., & Ying, L. (2021). On linear stability of sgd and input-smoothness of neural networks. Advances in Neural Information Processing Systems, 34, 16805-16817.
   - Feng, Y., & Tu, Y. (2021). The inverse variance–flatness relation in stochastic gradient descent is critical for finding flat minima. Proceedings of the National Academy of Sciences, 118(9), e2015617118.

**Questions:**

Question:

- Can you elaborate here and in the manuscript about the relationship between the Lindsay et al 2019 results and yours. In the manuscript you indicate “We hypothesize that these effects are due to constraints on the error signal reaching the retina, rather than the feedforward bottlenecks themselves.”. However, the bottleneck in the feedforward network can impose a lower rank in the feedback due to the different structure of W. There has been some work on the relationship of these properties in the implicit regularization literature, specially relating SGD with convolutional kernel learning. It would be good for them to be included in the discussion of these results because it seems like architecture and gradient structure are highly correlated.
   - Gunasekar, S., Lee, J. D., Soudry, D., & Srebro, N. (2018). Implicit bias of gradient descent on linear convolutional networks. Advances in neural information processing systems, 31.
   - Ortega Caro, J., Ju, Y., Pyle, R., Dey, S., Brendel, W., Anselmi, F., & Patel, A. B. Translational Symmetry in Convolutions with Localized Kernels Causes an Implicit Bias towards High Frequency Adversarial Examples. Frontiers in Computational Neuroscience, 18, 1387077.

Suggestions:

- Given the references above, It would be good to include these references because there are indications that SGD already works in a low dimensional space. In addition,  change the paragraph in the discussion about rethinking of gradient descent dynamics due to these references.

---

> ### Author Response · Authors · 2024-11-19
>
> We thank you for your comments and for taking the time to consider our work. We are pleased you appreciated the relationship between the error feedback and the receptive fields. This question originally prompted our study.
>
> **Response to the mentioned weaknesses**
> 1. You correctly noted that our datasets are relatively simple. However, we do not think a more complex dataset will bring further insight. **The main message of our work is that if the output error is low-dimensional, then we can train large errors with low-dimensional error feedback if the feedback weights correctly learn the error space**. Using image data, we show that this principle extends to nonlinear and convolutional networks and that the error signal shapes the receptive fields. Even training on ImageNet-1k would require a feedback rank of 1000. If successful, which we believe it will be, it will not highlight the effectiveness of low-dimensional feedback. Instead, we elucidate our conclusions that deep networks can learn efficiently using minimal error signals. This is an important finding for studying brain-circuit architecture but not an alternative solution for learning complex visual data.
>
>     Finally, **your comment raises a fascinating question: can a network learn a complex dataset, such as ImageNet, with an error signal with a lower output dimensionality?** We imagine the top layer would see the full 20k class error feedback, but deeper layers would only see a low-dimensional error, which will be learned. It may be possible due to the structure of the data and labels. However, this question is outside the scope of this work. Nevertheless, we are training a network on ImageNet and will report here if we have interesting findings.
>
> 2. Novelty of low-rank gradients. Indeed, there has been work showing that the dynamics of weights during SGD are low dimensional. **This aligns with our hypothesis that a low-dimensional error signal is potentially sufficient for training**. We have added these references to our introduction. However, previous studies on regularization do not address whether a low-dimensional error can drive the weight dynamics. Regularization and the shape of the loss-landscape [Ma and Ying 2021, Feng and Tu 2021] can constrain dimensionality, but it is not equivalent to a restricted indirect error channel; the latter is of greater importance to neuroscience. Finally, previous studies on the dimensionality of training dynamics do not systematically study the relation between the error signal, the performance, and the receptive fields.
>
> **References**
> - Ma, C., & Ying, L. (2021). On linear stability of sgd and input-smoothness of neural networks. *Advances in Neural Information Processing Systems*, *34*, 16805-16817.
> - Feng, Y., & Tu, Y. (2021). The inverse variance–flatness relation in stochastic gradient descent is critical for finding flat minima. *Proceedings of the National Academy of Sciences*

---

> ### Author Response · Authors · 2024-11-19
>
> **An answer to the specific question**
>
> In [Lindsey et al. 2019], the authors observed that network bottlenecks result in a systematic change in the receptive fields, with tighter bottlenecks resulting in more center-surround fields. We find this a fascinating result. Not because of the direct biological implications described by Lindsey et al.—the retinal layer probably does not learn in a supervised manner—but because it relates architecture to receptive fields, offering a powerful tool for computational neuroscience. However, they did not study the mechanism underlying the emergence of the center-surround fields. Moreover, their results raise a conundrum: a bottleneck in the feedforward pass should require more efficient coding, while center-surround is less efficient than Gabor-like filters.
> **The proposition that it is due to the restricted error pathway is perhaps intuitive, but our work is the first time this can be systematically studied**. Indeed, we find that restricting the error is sufficient to shape the receptive fields (Fig 5). Furthermore, in the appendix, we show that a network with a bottleneck—similar to that of Lindsey et al.—but with a “full-rank” error that bypasses the bottleneck does not lead to center-surround filters. Our results hold further implications for computational neuroscience;  we show that the shape, particularly the symmetry, of tuning curves is affected by the available error signal.
> We note that restricting the error dimensionality is different from regularizing it. Gunasekar’s work studies the dimensionality of the solution and does not restrict the error during learning. Similarly, Caro et al.'s work studies symmetries in the feedforward and the solution but not the independent effects of the backward pass.
>
> **References**
>
> - Gunasekar, S., Lee, J. D., Soudry, D., & Srebro, N. (2018). Implicit bias of gradient descent on linear convolutional networks. *Advances in neural information processing systems*, *31*.
> - Caro, J. O., Ju, Y., Pyle, R., Dey, S., Brendel, W., Anselmi, F., & Patel, A. B. (2024). Translational symmetry in convolutions with localized kernels causes an implicit bias toward high frequency adversarial examples. *Frontiers in Computational Neuroscience*, *18*.

---

### Author Response · Authors · 2024-11-27
**Summary of revisions in the manuscript**

We thank all the reviewers for their comments, stimulating discussions, and mostly for their time thinking about our work. We’ve uploaded a revised version of the paper, which resolves specific questions and comments raised in the discussion. In particular, we have implemented the following changes:

1. We’ve **added a baseline for the retinal receptive fields** trained with backpropagation and a bottleneck (as in reference [Lindsey et al. 2019]). The results appear in the appendix together with the results of training a network with bottleneck feedforward but a full-rank backward. Together with Fig 5, they further support the claim that the emergence of center-surround receptive fields is due to the limited error pathway.
2. We **added all references brought up in the discussion** with the appropriate context, as appears in the discussions above. In particular, we acknowledged previous works that studied the dimensionality of the weight update and its relation to two symmetries in connectivity and regularization. We have emphasized that none of the previous studies the possibility and effects of constraining the error signal.
3. We have emphasized the novelty of this work with respect to previous learning studies of Feedback Alignment, in particular [Akrout 2019]. We explain, in the background section, that **not only did the previous study not consider a constrained feedback matrix, but they failed in this case** (as we show in Section 3).
4. In our conclusion, we have emphasized the main conceptual novelty in our results: that **when the error signal is constrained, the feedback must align not only with the feedforward weights (the feedback matrix column space), but also with the error (the feedback matrix row space).**
5. We **replaced the ambiguous term “biologically plausible”** with a more direct and detailed description of what we are doing.
6. We increased the font size in the figure and corrected typos throughout.

---

### Meta-Review · Area_Chair_mgf5 · 2024-12-25

**Metareview:**

The paper introduces Restricted Adaptive Feedback (RAF), a novel learning rule that leverages low-dimensional error feedback to train neural networks, challenging the traditional high-dimensional backpropagation paradigm. The authors argue that low-dimensional error signals can achieve performance comparable to standard methods, providing theoretical analysis for linear networks and extending their ideas to nonlinear and convolutional architectures.

The reviewers appreciate the promising ideas, interesting empirical findings, and theoretical foundation provided in the paper. Meanwhile, there are concerns about the limited experimental scope, discussion of prior work, and the mechanism in nonlinear networks. The authors are encouraged to take into account these comments in the next version of this paper.

**Additional Comments On Reviewer Discussion:**

The paper received mixed opinions. The reviewers found the proposed approach very interesting and promising, but also felt that the experimental and theoretical support could be enhanced. After the rebuttal period, a few concerns remained: (i) Limited experimental scope. Reviewers afy6 and QAwR gave concrete suggestions on expanding the experiment component of the paper, but the authors didn't make an update in the rebuttal period. (ii) Theoretical insights need to be extended especially for non-linear networks. (iii) Some claims about related work are not entirely accurate.

---

### Decision · Program_Chairs · 2025-01-22

Reject